# Systematic investigation of chemo-immunotherapy synergism to shift anti-PD-1 resistance in cancer

Yue Wang[1,4], Dhamotharan Pattarayan [1,4], Haozhe Huang[1,4], Yueshan Zhao[1], Sihan Li[1], Yifei Wang [1], Min Zhang[1], Song Li [1] & Da Yang [1,2,3] ✉

Chemo-immunotherapy combinations have been regarded as one of the most practical ways to improve immunotherapy response in cancer patients. In this study, we integrate the transcriptomics data from anti-PD-1-treated tumors and compound-treated cancer cell lines to systematically screen for chemo-immunotherapy synergisms in silico. Through analyzing anti-PD-1 induced expression changes in patient tumors, we develop a shift ability score to measure if a chemotherapy or a small molecule inhibitor treatment can shift anti-PD-1 resistance in tumor cells. By applying shift ability analysis to 41,321 compounds and 16,853 shRNA treated cancer cell lines transcriptomic data, we characterize the landscape of chemo-immunotherapy synergism and experimentally validated a mitochondrial RNA-dependent mechanism for drug-induced immune activation in tumor. Our study represents an effort to mechanistically characterize chemo-immunotherapy synergism and will facilitate future pre-clinical and clinical studies.

Inhibitory immune checkpoints, including programmed death 1 (PD-1), negatively regulate the cytolytic activities of cytotoxic T cells via interactions with their ligands on tumor cells. Blockage of these interactions can restore anti-tumor immune response of T cells and prevent tumor immune surveillance[1]. Immunotherapy using PD-1 blockade, a.k.a. anti-PD-1, has significantly improved patient prognosis in different cancer types such as melanoma[2,3], lung cancer[4], colorectal cancer[5], and triple-negative breast cancer[6]. However, anti-PD-1 therapy is still not available for majority of cancer patients. Studies showed that the response rate of anti-PD-1 in melanoma patients ranges from 20 to 30%[2,3]. In other cancer types such as breast cancer, prostate cancer, and colorectal cancers, the anti-PD-1 response rates range from 13 to 38%[7]. Even for patients who initially respond to the therapy, the later developed drug resistance remains to be challenging[2,8]. There is an urgent need to identify effective strategies to overcome anti-PD-1 resistance and improve the overall response rate.

Emerging studies have reported that some chemo- and targeted therapy agents can induce significant effects on immune response in tumors[9]. For example, gemcitabine is a synthetic pyrimidine nucleoside analog which has been widely used as standard-of-care treatments in various cancers[10,11]. Gemcitabine can induce immunogenic cell death, which enhance the dendritic cell-dependent cross-presentation of tumor antigens to cytotoxic T cells[12,13]. Of note, by 2023, FDA have approved many chemo-immunotherapy regimens in diffuse large B-cell lymphoma (Polatuzumab + bendamustine/rituximab), triple-negative breast cancer (Atezolizumab/Pembrolizumab + taxanes), gastric cancer and esophageal adenocarcinoma (Nivolumab + FU-/platinum)[14]. As more chemo-immunotherapy combination regimens are being investigated and validated by ongoing clinical trials[15,16], they are becoming one of the most feasible paths to obtaining durable, long-lasting immunotherapy responses.

However, the design of the combination regimens so far is largely relied on clinical experiences, it is very challenging to characterize new chemo-immunotherapy synergisms[14,17]. The emerging large-scale pharmacological transcriptomic datasets that profile the expression changes after drug/immunotherapy treatment provide deeper and

[1]Center for Pharmacogenetics, Department of Pharmaceutical Sciences, University of Pittsburgh, Pittsburgh, PA 15261, USA. [2]UPMC Hillman Cancer Institute, University of Pittsburgh, Pittsburgh, PA 15261, USA. [3]Department of Computational and Systems Biology, University of Pittsburgh, Pittsburgh, PA 15261, USA. [4]These authors contributed equally: Yue Wang, Dhamotharan Pattarayan, Haozhe Huang. ✉e-mail: dyang@pitt.edu

novel insights on how treatment changes biological processes in tumor[18]. These data present us an excellent opportunity to computationally model the interaction between chemotherapy and immunotherapy.

In this study, we hypothesize that the treatment-induced gene expression changes in tumor could be utilized to determine the anti-PD-1 therapy outcome and to reveal the underlying resistance mechanism. Using anti-PD-1 induced expression changes, we characterize gene signatures that are robustly associated with immune checkpoint blockade responses in patients. Importantly, we demonstrate that genetic inhibition of these signature genes can shift the anti-PD-1 response phenotypes. With these observations, we develop the shift ability score to quantify a treatment's capability of improving anti-PD-1 response. Through in silico screening on 41,321 compound-treated and 16,853 shRNA-treated cell line expression profiles, we identify treatments that can potentially shift anti-PD-1 resistance. Finally, we reveal that a mitochondrial RNA-dependent activation of type I interferon signaling may be a promising mechanism for chemo-immunotherapy synergism.

## Results

### Robust treatment-induced expression changes associated with patient anti-PD-1 response

To identify genes associated with acquired anti-PD-1 resistance in patients, we obtained pre-post-treatment paired transcriptomic data of tumor biopsies from melanoma patients who received anti-PD-1 therapy (GSE91061)[19] (Supplementary Fig. 1a and Supplementary Data 1). Through principal component analysis (PCA), we observed that most of the variances between anti-PD-1 responders and non-responders can be explained by treatment-induced expression (AUC = 0.77) but not treatment-naïve expression (AUC = 0.55) (Fig. 1a). It appears to us that treatment-induced expression changes can better characterize the underlying mechanisms of anti-PD-1 resistance in patients.

In this regard, we sought to identify treatment-induced expression changes that robustly associate with anti-PD-1 response. We implemented a bootstrapping and cross-validation feature selection procedure, leading to the identification of 419 genes for anti-PD-1 resistance (R) signature and 366 genes for anti-PD-1 sensitivity (S)

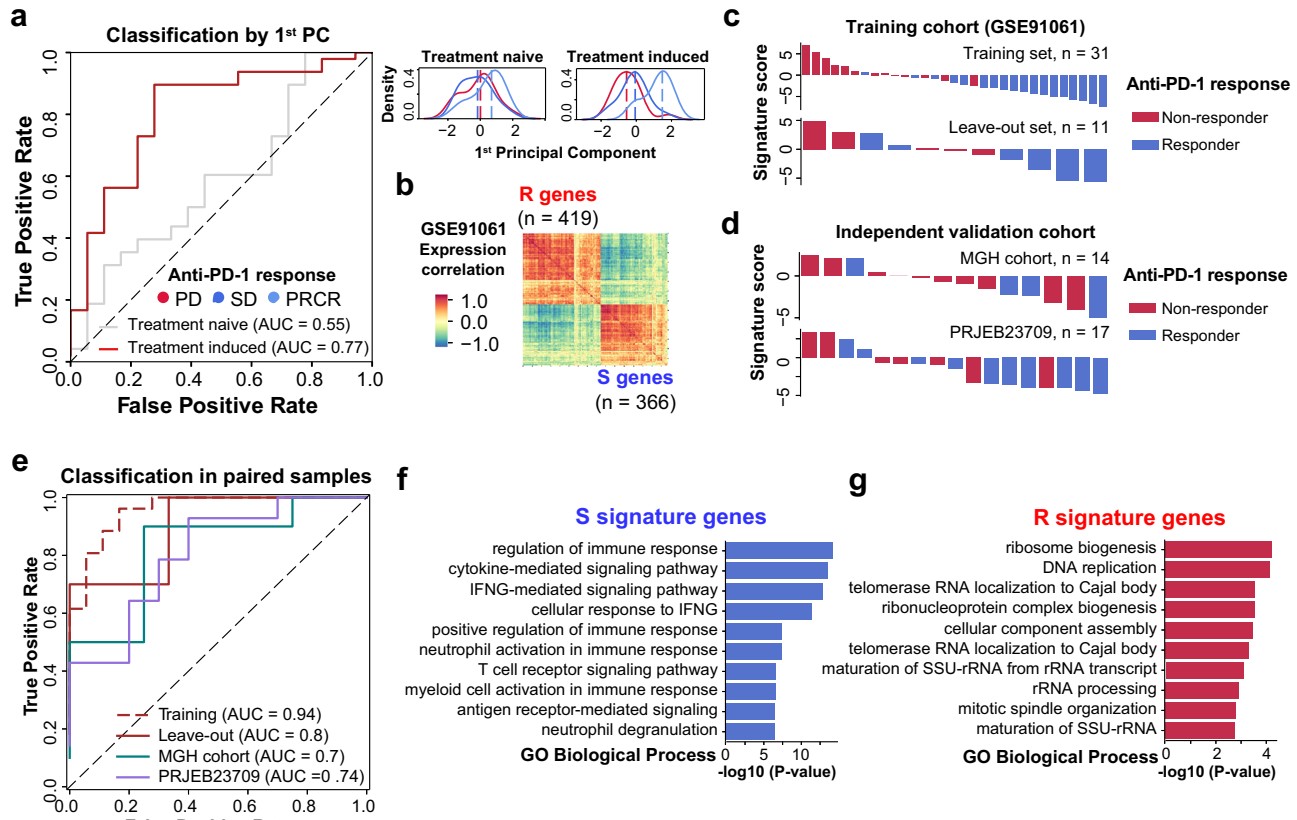

**Fig. 1 | Robust treatment-induced expression changes associated with anti-PD-1 response in melanoma patients. a** Receiver operating characteristic (ROC) curve showing the performance of using treatment-naïve (gray) or treatment-induced (red) expression to classify anti-PD-1 responders and non-responders. The kernel density estimation plot shows the distribution of patient response groups on the first principal component of treatment-naïve expression (upper) or treatment-induced expression (lower) (n = 42). Source Data are provided as Supplementary Data 1. **b** Expression correlation between 419 Resistance signature genes and 366 Sensitivity signature genes in melanoma patients (n = 42). Colormap represents the correlation coefficient given by Pearson's correlation. Source Data are provided as Supplementary Data 1. **c** Integrating R and S signature to classify anti-PD-1 responders and non-responders in training cohort (GSE91061). Patients are ranked in descending order based on signature score, which is given by the difference of enrichment score between S signature and R signature. Colors of the bar indicate the anti-PD-1 response group. Source Data are provided as Supplementary Data 1. **d** Validation of R and S signature in two independent validation cohorts. Patients are ranked in descending order based on signature score, which is given by the difference of enrichment score between S signature and R signature. Colors of the bar indicate the anti-PD-1 response group. Source Data are provided as Supplementary Data 1. **e** Receiver operating characteristic (ROC) curve summarizing the performance of using R and S signatures to classify anti-PD-1 responders and non-responders. Training set, n = 31; Leave-out validation set, n = 11; MGH cohort, n = 14; PRJEB23709 cohort, n = 17. Source Data are provided as Supplementary Data 1. **f** GO Biological Process: Pathway enrichment of genes involved in S signature. X-axis represents adjusted P value derived from gene set enrichment analysis. The enrichment P value is given by the "enrichr" function in GSEA. **g** GO Biological Process: Pathway enrichment of genes involved in R signature. X-axis represents adjusted P value derived from gene set enrichment analysis. The enrichment P value is given by the "enrichr" function in GSEA.

signature (see *Methods*; Supplementary Fig. 1b, c and Supplementary Data 1). Genes involved in R signature are generally anti-correlated with genes in S signature (Fig. 1b). Combining the treatment-induced expression changes of R and S signature can precisely recapitulate patient responses to anti-PD-1 treatment (cross-validation AUC = 0.94, leave-out validation AUC = 0.8; Fig. 1c, e). The performance of R and S signature is further validated by two independent melanoma cohorts who received anti-PD-1 treatment and had paired pre-post-treatment samples available (PRJEB23709 AUC = 0.74, MGH cohort AUC = 0.7)[20] (Fig. 1d, e). For patients from study which does not have paired pre-post-treatment samples (PHS001919)[21], the R and S signature achieved better performance in classifying anti-PD-1 responses in post-treatment cohorts (AUC = 0.74) than in pre-treatment cohorts (AUC = 0.44) (Supplementary Fig. 1d and Supplementary Data 1).

We noticed that S signature genes are highly enriched in pathways related to anti-cancer immunity (Fig. 1f and Supplementary Fig. 1f), while R signature genes are more enriched in immune evasion and cancer progression (Fig. 1g and Supplementary Fig. 1f). We further observed that the anti-correlation between R and S genes not only shows in anti-PD-1 treated patients ($R = -0.56$, $P = 0.0$, Pearson's correlation) but also shows in tumors from The Cancer Genome Atlas (TCGA) patient cohorts ($R = -0.53$, $P = 0.0$; Supplementary Fig. 1g, h). When looking into the correlation between the expression of R and S signatures with immune cell infiltration in TCGA patients (see *Methods*), we observed that S signature is strongly correlated with immune-hot phenotypes in multiple cancer types in addition to melanoma (Supplementary Fig. 1i and Supplementary Data 2), whereas increased expression of R signature is generally correlated with immune-suppressive microenvironment (Supplementary Fig. 1j and Supplementary Data 2). In a cohort of anti-PD-1 treated patients from multiple cancer types, these two signatures still show classification capability for patient response in squamous lung cancer (AUC = 0.72) and non-squamous lung cancer (AUC = 0.68) even when only pre-treatment expression is available (Supplementary Fig. 1k).

### Genetic inhibition of genes in R and S signature can shift anti-PD-1 response phenotypes

Given the robust association between anti-PD-1 response and the expression of signatures genes in patient samples, we wondered if some of the signature genes can regulate anti-cancer immune response. In this regard, we integrated the post-shRNA-treatment transcriptomes of 10 cancer cell lines across 6 cancer types from Connectivity Map 2020 (CMAP2020)[18](Fig. 2a). Of 3488 shRNA targeted genes, 257 of them are involved in R or S signature (Fig. 2b, Supplementary Fig. 2a, b and Supplementary Data 3). We designed a metric named "shift ability score" to quantify if an shRNA treatment can simultaneously suppress R signature genes and induce S signature genes, and vice versa (Fig. 2c; see *Methods*).

Interestingly, we found R-targeted shRNAs (shRs) and S-targeted shRNAs (shSs) showed an opposite direction of shift ability (Fig. 2c). Among the shRs that can successfully inhibit target gene expression, 73% can suppress the overall R signature expression in the same cell line. Surprisingly, 90% of these shRs can upregulate the S signature expression at the same time, indicating a resistant-to-sensitive (R-to-S) shifting (Fig. 2c, d and Supplementary Fig. 2c). On the other hand, 68% of the shSs can successfully inhibit target gene expression, of which 51% can suppress the overall S signature expression in the same cell line. Similar to the observations in shRs, 75% of these shSs can simultaneously increase the overall expression of R signature, indicating a sensitive-to-resistant (S-to-R) shifting (Fig. 2c, d and Supplementary Fig. 2d). Notably, among the R genes whose genetic inhibition resulted in R-to-S shifting, many of them have been reported as master regulators of tumor immune response, including MYC[22], PTK2[23], BIRC5[24] and CDK2[25] (Fig. 2e). These observations strongly suggest an underlying causal and functional relationship, rather than simple correlations, between the signature genes and immunotherapy response.

Building upon these findings, we performed an in silico screening using the entire shRNA libraries (Fig. 2f) and identified 546 potential genes whose genetic inhibition would induce significant R-to-S shifting in at least one tested cell line (Fig. 2g, h). The identified genes recapitulated several promising targets that have been established to help overcome anti-PD-1 resistance. Knockdown of these genes showed a significant suppression of R signature and increase of S signature (Supplementary Fig. 2e). These genes also showed a higher treatment-induced expression changes in anti-PD-1 resistant patients (Fig. 2i). For instance, studies have reported increased VDAC1 expression may drive dysregulated anti-tumor immunity, and silencing of VDAC could help reprogramming tumor microenvironment[26–28]. RRM1, the target of ribonucleotide reductase inhibitors, also exhibited significant R-to-S shifting potential in multiple cell lines after being genetically inhibited. This is consistent with the clinical application that ribonucleotide reductase inhibitors, e.g., gemcitabine and fludarabine, can effectively synergize with PD-1 blockade and reverse anti-PD-1 resistance[29–31]. Taken together, shift ability analysis, which is based on treatment-induced changes on R signature and S signature, can be used to screen treatment that will potentially synergize with anti-PD-1 therapy and shift anti-PD-1 resistance.

### Shift ability analysis on post-compound-treatment transcriptomes characterized chemo-immunotherapy synergism

Next, we sought to apply shift ability analysis to characterize compounds that can synergize with anti-PD-1 treatment. We collected 41,321 post-treatment transcriptome profiles across 64 cell lines from CMAP2020 database. These cell lines were treated by 4264 compounds targeting 392 pathways at different dosages. By evaluating the shift ability of each treatment experiment, we finally identified 948 R-to-S shifting compounds who showed significant R-to-S shift in at least one experiment across cell lines (see *Methods*; Supplementary Data 4).

Among the identified compounds, we found some of them exhibited a cancer-specific enrichment mechanism of actions (Fig. 3a, b and Supplementary Fig. 3a). For example, MEK inhibitors are ranked high for R-to-S shifting ability in A375, A549, HCC515, HELA and YAPC cells. This is consistent with the clinical indication that MEK signaling is activated in melanoma, colorectal cancer, non-small cell lung cancer, ovarian cancer, and pancreatic cancer[32–34]. The Estrogen receptor antagonist, on the other hand, showed significant R-to-S shifting ability exclusively in ER$^+$ cell lines (MCF7 and VCAP), indicating its shift ability is relied upon cell lines' expression of estrogen receptor. Another example is EGFR inhibitors, who showed significantly higher R-to-S shifting in EGFR expressing HCC515 compared to other cell lines[35]. These results suggest that, for those small molecule inhibitors, the induction of the R-to-S shift partially depends on the accessibility of drug targets.

Interestingly, we also observed that many drugs can induce R-to-S shifting in "pan-cancer" manner (Fig. 3c). For example, mitoxantrone, a topoisomerase inhibitor, showed significant R-to-S shifting ability in 11 cell lines across multiple cancer types (Fig. 3c, d and Supplementary Fig. 3b). This observation is consistent with previous findings that, in many cancer types, mitoxantrone can induce immunogenic cell death, which will activate type I interferon signaling and facilitate the MHC-II-mediated antigen presentation through dendritic cells[36–38]. Other topoisomerase inhibitors, including DXR (Supplementary Data 4), also showed R-to-S shifting potential in multiple cell lines.

We next validated the synergism between DXR, and anti-PD-1 therapy in melanoma (B16), prostate cancer (MyC-CaP) and colorectal cancer (CT26) syngeneic mouse model in vivo (Fig. 3e, f). Further flow cytometry analysis of tumor-infiltrating lymphocytes revealed significantly altered tumor microenvironment after treatment (Fig. 3g, h and Supplementary Fig. 3d–h). Particularly, combination treatment of DXR and anti-PD-1 treatment significantly increased CD4$^+$

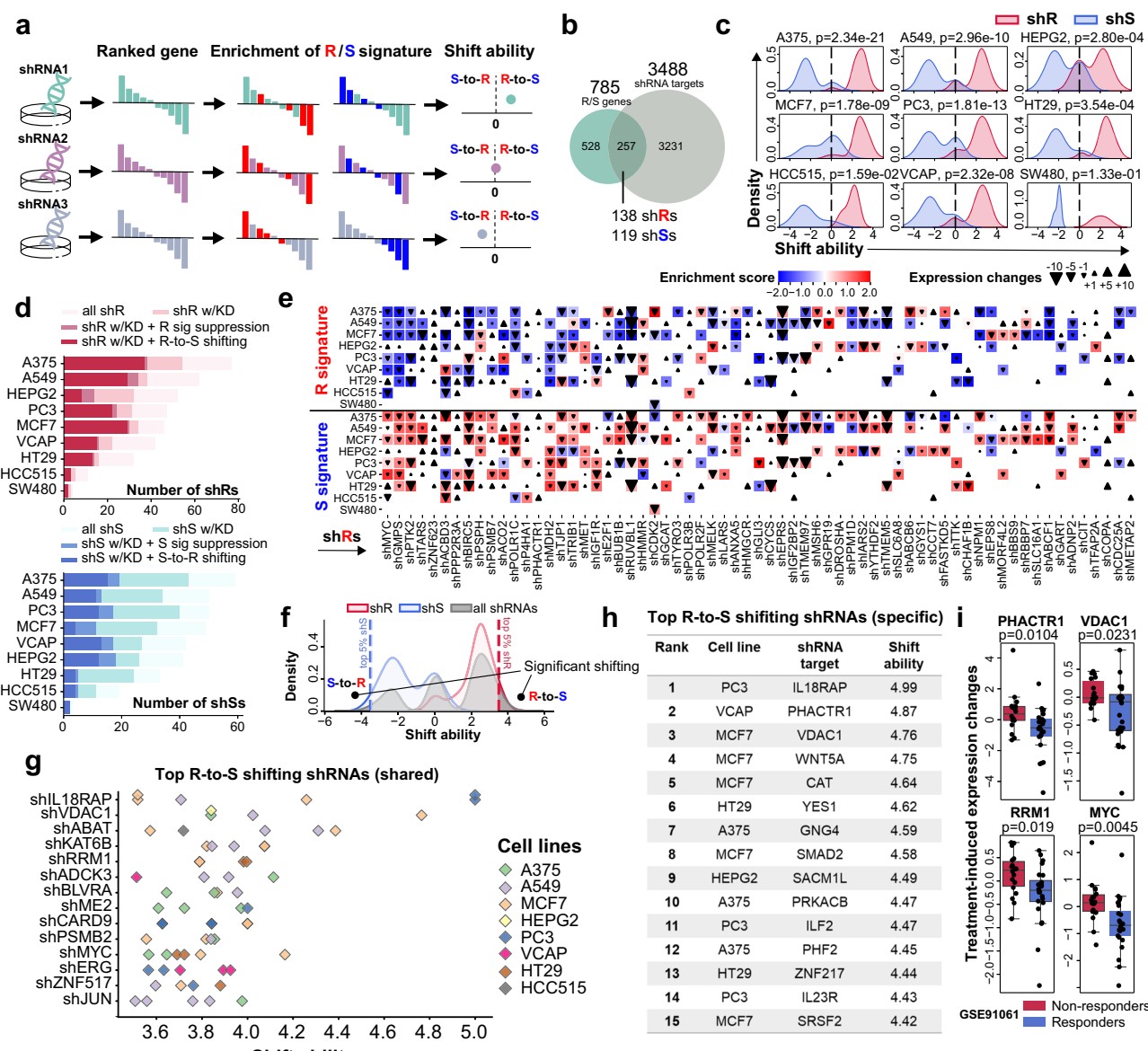

**Fig. 2 | Genetic inhibition of genes in R and S signature can shift immunotherapy response phenotypes. a** Graph demonstration of R and S signature enrichment and shift ability analysis. **b** Number of R genes and S genes that are being targeted by shRNAs in Connectivity Map. **c** Distribution of shift ability score of shRNAs targeting R signatures (shR) or S signatures (shS) across different cell lines. *P* values (two-sided) are given by two sample KS test. Source Data are provided as Supplementary Data 3. **d** Number of R (top) or S (bottom) targeting shRNAs that are able to knock down the target genes (shX w/KD), to suppress the target signature (X sig suppression), and to induce R (S) to S (R) shifting. Source Data are provided as Supplementary Data 3. **e** Suppression of R signature (above) and induction of S signature (bottom) by R-targeting shRNAs. Colormap indicates the normalized enrichment score of corresponding signatures in each experiment. The size of triangles represents the shRNA-induced target gene expression changes

compared to other experiments in the same panel. Direction of triangles indicates the direction of expression changes. Source Data are provided as Supplementary Data 3. **f** Definition of significant shifting based on the shift ability distribution of signature-targeting shRNAs. **g** Top R-to-S shifting shRNAs that shared across multiple cell lines. Source Data are provided as Supplementary Data 3. **h** List of shRNAs with highest R-to-S shifting ability. Source Data are provided as Supplementary Data 3. **i** Treatment-induced expression changes of selected shRNA target genes in anti-PD-1 treated patient cohorts (non-responders, $n = 18$; responders, $n = 24$). *P* values (two-sided) are given by two sample student's t test. Center lines represent median treatment-induced expression changes, the box limit indicates the lower quantile and upper quantile, and whiskers represent the minimal and maximal treatment-induced expression changes.

---

IFN-γ⁺ and significantly decreased M2 macrophage populations in the tumor microenvironment (Fig. 3g, h and Supplementary Fig. 3d–h). These results demonstrated that DXR treatment can active tumor immune response and is synergistic with anti-PD-1.

### Integrating shRNA and compound screening identifies PAK4 as a potent target for chemo-immunotherapy synergism

To further characterize targets for chemo-immunotherapy synergisms, we integrated the compound (Supplementary Data 4) and

shRNA (Supplementary Data 5) screening result and identified 14 potent synergism targets. The genetic and pharmacological inhibition of these genes can induce consistent R-to-S shifting in the same cell lines. Their expression is also strongly associated with suppressed anti-tumor immunity in patient tumors across 32 TCGA cancer types (Fig. 4a and Supplementary Data 5).

Of note, the prioritized drug targets not only included previously reported chemoimmunotherapy synergisms, such as BRAF[39], RRM1[12,13], CDK1[40], CDK2[41], HDAC2[42], but also a couple of targets whose capability

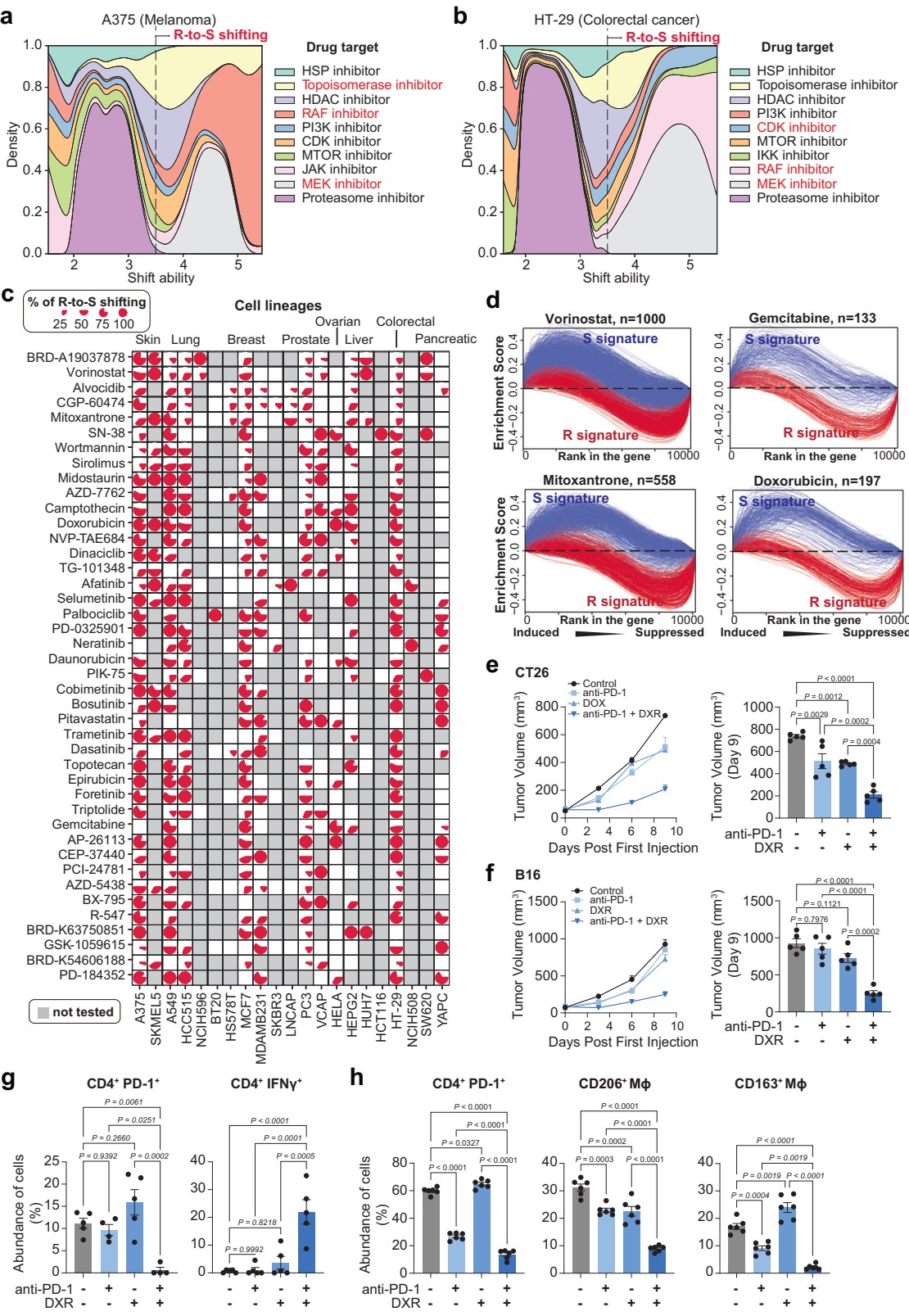

of regulating immune response have not been studied until recent. For example, both genetic knockdown and pharmacological inhibition of PAK4 showed drastic R-to-S shift ability in A375 (melanoma) (Fig. 4b, c). For cell lines in which only PAK inhibitor (i.e., PF-03758309) data are available, we also observed significant R-to-S shift ability induced by PAKi in MCF7 (breast cancer), PC3 (prostate cancer) and HCC515 (lung cancer) (Supplementary Fig. 4a–e).

In patient tumor samples, PAK4's inhibition is positively correlated with immune-hot tumor microenvironment in 22 cancer types. Among which breast cancer, kidney cancer, prostate cancer, melanoma, and colorectal cancer showed the most significant correlation (Fig. 4d and Supplementary Fig. 4f–h). In three independent cohorts of patients treated by anti-PD-1 therapy, PAK4 shows higher treatment-induced expression changes in non-responders than in responders

**Fig. 3 | Shift ability analysis on compound-treated transcriptomes identified the landscape of chemo-immunotherapy synergism. a** Stacked density plot of top R-to-S shifting drug targets in A375 melanoma cell line. X-axis indicates shift ability. The Y-axis indicates density. Red-highlighted text indicates the major drug targets in significant R-to-S shifting range (shift ability >= 3.5). Source Data are provided as Supplementary Data 4. **b** Stacked density plot of top R-to-S shifting drug targets in HT29 colorectal cell line. X-axis indicates shift ability. The Y-axis indicates density. Red-highlighted text indicates the major drug targets in significant R-to-S shifting range (shift ability >= 3.5). Source Data are provided as Supplementary Data 4. **c** Compounds that showed R-to-S shifting in multiple cell lines. Pie charts in each cell indicate the percentage of experiments showed a R-to-S shift ability. The bar plot on the right side of the pie matrix indicates the number of cell lines where the compounds showed R-to-S shifting in at least one experiment. Untested cell lines are shaded by gray. Source Data are provided as Supplementary

Data 4. **d** Enrichment curves of R signature and S signature in vorinostat, gemcitabine, mitoxantrone or doxorubicin treated cell lines. **e** CT26 tumor volume ($n = 5$ mice) changes and tumor volume on Day 9 in mice following treatment with anti-PD-1, doxorubicin (DXR) and combination of anti-PD-1 with DXR. **f** B16 tumor volume ($n = 5$ mice) changes and tumor volume on Day 9 in mice following treatment with anti-PD-1, DXR and combination of anti-PD-1 with DXR. **g** Single-cell suspensions were prepared from B16 melanoma samples ($n = 5$ mice) and subjected to flow cytometry analysis including CD4$^+$ PD-1$^+$ T cells (left panel) and CD4$^+$ IFNγ$^+$ T cells (right panel). **h** MyC-CaP prostate cancer samples ($n = 6$ mice) infiltrated immune cells analysis including CD4$^+$ PD-1$^+$ T cells (left panel), CD206$^+$ macrophages (middle panel) and CD163$^+$ macrophages (right panel). Data in (**e**–**h**) are presented as mean ± SEM, P values were generated using one-way ANOVA with Tukey's post hoc test for comparison.

(Fig. 4e). These observations reinforce the importance of including both pre- and post-treatment profiling in identifying key regulators of anti-PD-1 response.

Indeed, recent studies have shown that pharmacological inhibition of PAK4 is a very promising therapy to be combined with immunotherapies. This includes compound PF-03758309, which can reprogram vascular microenvironments and improve CAR-T therapy in glioblastoma[43], as well as compound KPT-9274, which can increase T cell infiltration and improve anti-PD-1 response in melanoma[21]. However, it is not clear how targeting PAK4 can boost chemo-immunotherapy synergism.

### The treatment-induced mitochondria damage may be a mechanism for chemo-immunotherapy synergism

We next sought to systematically characterize the potential mechanism(s) for the synergism between anti-PD-1 and R-to-S shifting compounds. Overall, 948 R-to-S shifting compounds can be clustered into two groups based on treatment-induced changes of different molecular processes (Fig. 5a). The most common activated molecular processes from each cluster revealed that one major cluster ("C-immune") showed a direct induction of immune response (NES = 2.25, FDR < 1e-3). The compound in this cluster appears to be able to direct induce the genes involved in antigen presentation and immune cell recruitment[44]. In contrast, the other major cluster ("C-stimulus") exhibited a significant induction of type I interferon (NES = 2.25, FDR < 1e-4), suggesting the compounds triggered some stimulus, which further activate the interferon pathways (Fig. 5a and Supplementary Data 6).

Most compounds in cluster "C-stimulus" seems to induce mitochondria damage related processes, including mitophagy (autophagy of mitochondrion, NES = 2.17, FDR < 1e-2). Mitophagy is an essential cellular process that tumor cells rely on to deal with the damaged mitochondria and protect themselves from chemotherapy-induced cell death[45]. Recently, mitochondrial DNA (mtDNA) and RNA (mtRNA) released from damaged mitochondria have been characterized as triggers of tumor-intrinsic immune response[46–48]. For example, DXR, which has been demonstrated to cause mitochondrial damage[49], is clustered as a "C-stimulus" drug (Fig. 5a and Supplementary Data 6). Using a specific mitophagy detection assay (see *Methods*), we have shown that DXR treatment can increase mitophagy activation in MCF7 and A549 cancer cells in a dose-dependent manner (Supplementary Fig. 5a–c). This observation suggests DXR induced mitochondria damage may be a mechanism for its synergy with anti-PD-1 treatment.

Notably, PF-03758309 (PAKi) is also categorized to be one of "C-stimulus" drugs. Although PAKi have been shown to have strong synergism with anti-PD-1 in previous studies[21,43], the underlying mechanism remains elusive. Our analysis revealed that PAKi can strongly activate pathways relevant to mitochondrial instability (Supplementary Fig. 5d). To validate our computational analysis, we treated various cancer cells with PAKi and observed that PAKi treatment

increases LC3-I and LC3-II protein levels (Fig. 5b and Supplementary Fig. 5e) and mitophagy[50,51] (Fig. 5c, d and Supplementary Fig. 5f).

### PAKi treatment activates type I interferon signaling through increasing cytosolic presence of mtRNAs

Increased mitophagy activation is a strong indication of mitochondrial damage caused by PAKi treatment. Mitochondrial damage results in efflux of mtRNA and mtDNA into cytoplasm[52], which can activate MAVS or cGAS-STING mediated immune responses[46,53,54]. Both transcriptomics analysis and qPCR validation reveal that PAKi treatment induces a dose-dependent upregulation of antigen presentation genes in multiple cancer cell lines (Fig. 5a, e, f, Supplementary Fig. 5g and Supplementary Tables 1, 2). Furthermore, PAKi treatment can upregulate type I interferon genes, downstream targets of interferon stimulated genes (Fig. 5g, Supplementary Fig. 5d, g, h and Supplementary Tables 3, 4), CXCL10 (Fig. 5h, i), and *PD-L1* expression in multiple cancer cells in a dose-dependent manner (Fig. 5j).

We first investigated if PAKi treatment's impact is mediated by the mtDNA-STING pathway. Interestingly, PAKi treatment did not lead to STING activation (i.e., phosphorylated STING) or expression (i.e., total STING) (Supplementary Fig. 6a). Moreover, enzymatic DNA depletion cannot abolish PAKi-induced activation of type I interferon signaling (Supplementary Fig. 6b–d). These observations suggest that PAKi-induced interferon signaling is not mediated by mtDNA-STING signaling.

We next sought to determine if the cytosolic presence of mtRNAs mediates the immune response[52,54] (Fig. 6a). qPCR analysis of the cytosolic cell fractions reveals that PAKi treatment significantly increases the cytosolic concentration of mtRNAs (e.g., *MT-CO1*, *MT-ND5*, *MT-ND6*, and *MT-CYB*) (Fig. 6b, c and Supplementary Fig. 6e). Because of the bidirectional transcriptional activity of mitochondria, cytosolic mtRNAs can be immunogenic via forming double-stranded RNA (dsRNA)[54]. Indeed, immunofluorescence and flow cytometry analyzes using a dsRNA-specific monoclonal antibody (J2) shows that PAKi treatment increased cytosolic accumulation of dsRNA (Fig. 6d, e and Supplementary Fig. 6f, g). The aberrant cytosolic presence of dsRNA can be immunogenic through MAVS-dependent type I interferon pathway activation[53]. To further determine if PAKi-induced immune response depends on the dsRNA, we knocked out cytosolic dsRNA signaling protein MAVS and observed that the MAVS knockout (sgMAVS) significantly abolished type I interferon signaling, CXCL10, antigen presenting genes, and *PD-L1* expression induced by PAKi (Fig. 6f–h and Supplementary Fig. 6h). Notably, PAKi treatment consistently induced LC3 protein activation in both wild-type and MAVS knockout cells (Fig. 5f), suggesting the mitophagy itself on upstream of dsRNA-MAVS activation. Collectively, our results suggest that PAKi induces immune response in cancer cells through increasing cytosolic dsRNA that are released from mitochondria damage.

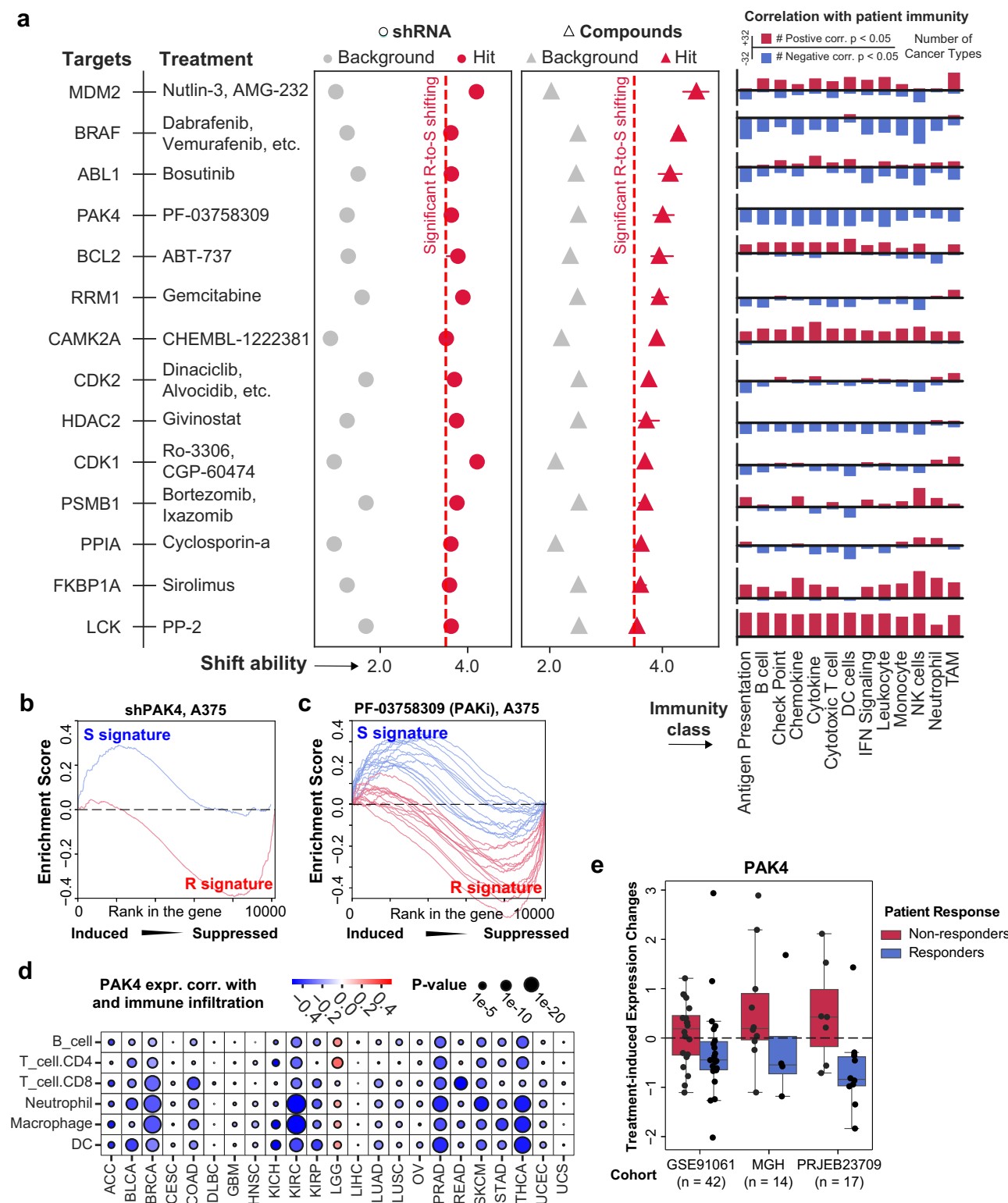

**b** shPAK4, A375

**c** PF-03758309 (PAKi), A375

**d** PAK4 expr. corr. with and immune infiltration

**e** PAK4

## Discussion

In this study, we have established the signatures of treatment-induced gene expression changes that are robustly associated with anti-PD-1 response. The association between the R and S signatures with patient response has been validated by multiple independent patient cohorts. Further functional analysis showed genes in these two signatures are highly associated with anti-tumor immune response in various cancer types. Notably, unlike other signature identification studies, which focused on predicting individual patient response in clinical situations,

our signatures are built to screen for promising chemo-drugs that can reverse anti-PD-1 resistance. Our analyses on shRNA-treated transcriptomic data demonstrated that a significant number of genes involved R and S signatures can functionally regulate anti-PD-1 response. These discoveries enlightened us to conceptualize the shift ability score and screen 4264 chemo-/targeted therapy compounds in multiple cancer types. By further integrating with the genetic knockdown screening, we identified gene targets whose pharmacological and genetic inhibition exhibit consistent

**Fig. 4 | Integrating shift ability analysis on genetic and pharmacological inhibition identified drug targets for chemo-immunotherapy synergism.**
**a** Prioritized drug targets for chemo-immunotherapy synergism. Drug names showed beside the gene targets are their corresponding pharmacological inhibitors. Circles indicate the shift ability of shRNAs. Triangles indicate the shift ability of compound treatment. Bar plots on the right side of the strip plot showed the number of TCGA cancer types where the corresponding genes have significantly positive (red) or negative (blue) correlation (Pearson's) with different anti-tumor immunity signatures. Source Data are provided as Supplementary Data 5. Enrichment curves of R signature and S signature in PAK4 knockdown (**b**) and PAK4 inhibitor treated (**c**) cell lines (A375). **d** Pearson's correlation between PAK4

expression and immune cell infiltration in TCGA samples (ACC, $n = 79$; BLCA, $n = 411$; BRCA, $n = 1097$; CESO, $n = 304$; COAD, $n = 467$; DLBC, $n = 48$; GBM, $n = 154$; HNSC, $n = 500$; KICH, $n = 65$; KIRP, $n = 288$; LGG, $n = 510$; LIHC, $n = 371$; LUAD, $n = 524$; LUSC, $n = 501$; OV, $n = 374$; PRAD, $n = 498$; READ, $n = 166$; SKCM, $n = 367$; STAD, $n = 375$; THCA, $n = 502$; UCEC, $n = 547$; UCS, $n = 56$). **e** Treatment-induced expression changes of PAK4 in patients before and after anti-PD-1 therapy (GSE91061, non-responders, $n = 18$, responders, $n = 24$; MGH cohort, non-responders, $n = 10$, responders, $n = 4$; PRJEB23709 cohort, non-responders, $n = 7$, responders, $n = 10$). Center lines represent median treatment-induced expression changes, the box limit indicates the lower quantile and upper quantile, and whiskers represent the minimal and maximal treatment-induced expression changes.

immunotherapy shift ability. We experimentally validated one FDA approved cancer drug, doxorubicin to be synergistic with the anti-PD-1 therapy in multiple mouse models. We expect these discoveries can be translated to patient care and have an impact on cancer therapy in the near future.

In addition to addressing the challenge of chemo-immunotherapy synergism, our shift ability analysis, grounded in treatment-induced expression changes, is also adaptable to various scenarios to eliminate unwanted consequences of treatment. For instance, with the availability of paired pre- and post-treatment transcriptomes, our signature construction procedure can be seamlessly applied to identify gene expression changes robustly associated with acquired resistance to chemo- or targeted therapies, drug-induced toxicity, or side effects such as aging[55]. The resulting signatures can then be applied to shift ability analysis, enabling a rapid in silico screening of compounds with the potential to reverse or eliminate the targeted condition.

On top of identifying chemo-immunotherapy synergism, the focus of drug-induced cancer cell transcriptomic data has also helped us to build a landscape on how drugs/compounds regulate cell-intrinsic mechanisms and eventually influence immunotherapy response. The tumor cell-intrinsic mechanisms are important to immunotherapy resistance[56]. Most of drugs that are FDA-proved to be combined with immunotherapy act through tumor-intrinsic immune activation, such as increasing tumor antigen presentation, immunogenetic cell death, and secretion of the cytokine[57]. Our study revealed two major mechanisms for the established chemo-immunotherapy synergisms. We found that some drugs, including FDA approved CDK inhibitors, can induce genes involved in the direct regulation of immune response. Other drugs, such gemcitabine, topoisomerase inhibitors, and MEK inhibitors, induce genes related to interferon response via activate some intrinsic stimuli, such as mitochondrial damage. We experimentally validated the PAK inhibitor PF-03758309 who can induce type I interferon through activating mtRNA-MAVS signaling in tumor. While drug-induced mitochondrial damage has been extensively studied to overcome chemo-resistance in cancer[45], its role in anti-tumor immunity has not been fully appreciated until recently[58–60]. Release of mtDNA and mtRNAs activates the anti-viral signaling, which will initiate the innate immune response[48,61]. Future in vitro and in vivo studies are warranted to determine whether drug-induced mitochondrial damage can be exploited to synergize immunotherapy.

The current study has limitations. Given the limited data availability of paired transcriptomes before and after anti-PD-1 therapy from other cancer types, the anti-PD-1 response signatures established in this study are based on melanoma patients. Although we have shown that our signatures are stable across various cancer types, the heterogeneity between cancer types should not be neglected for chemo-immunotherapy synergism identification. Therefore, we would like to emphasize that our in-silico shift ability screening may only serve as a pilot analysis, and experimental validation is strongly suggested for further synergism and mechanism investigation. In the future, when more paired data (pre- and post-treatment) are available for patients from other cancer types, we will be able to deliver cancer-specific signatures and design cancer-specific synergy screening procedures.

Collectively, our study has characterized a landscape for chemo-anti-PD-1 therapy synergism, which will facilitate the ongoing efforts on designing chemo-immunotherapy combinations to improve overall treatment outcomes in cancer patients.

## Methods
### Data collection and preprocessing
Post-perturbation cell line transcriptome data, including shRNA and compound treatment, were collected from the Expanded Connectivity Map (CMAP) LINCS Resource 2020 (complete version, 11/23/2021) through the CLUE portal (http://clue.io). The analyses were based on CMap level 5 signature matrices, with $238,351 \times 12,328$ in dimension for shRNA perturbation and $720,216 \times 12,328$ for compound treatment. Since the level 5 signatures were constructed based on replicates, only signatures with sufficient transcriptional activity score ($>= 0.4$) were retained for further analyses to reduce the false signals introduced by low reproducibility. Since CMap is based on L1000 panel which detected 978 landmark genes and then inferred the rest ten thousand genes based on the landmarks, we only utilized the expression information of 9196 best-inferred genes together with the landmark genes (10,174 genes in total). These filtering procedures led to the final matrix of $16,853 \times 10,174$ for shRNA perturbation and $41,321 \times 10,174$ for compound treatment.

Gene expression and clinical data of patients treated with immune checkpoint blockade were collected from Gene Expression Omnibus (GEO) with accession number GSE91061[62], GSE168204[62], GSE115821[63] and GSE93157[64], from European Nucleotide Archive (ENA) with accession number PRJEB23709[20], and from MTA transfer with University of California, Los Angeles (UCLA) with accession ID PHS001919[21].

Among them, GSE91061 has 43 unique pairs of melanoma samples ($n = 86$) pre- and on-treatment. According to the original publication, 18 pairs with progressive disease (PD) were considered as non-responders; 15 pairs with stable disease (SD) and 9 pairs with partial/complete response (PRCR) were considered as responders; 1 pair of samples with unknown (UNK) response was excluded from the analysis. GSE168204 and GSE115821 refers to the "MGH cohort" in the main text. There are in total 14 unique paired melanoma samples ($n = 28$) pre- and post-treatment, of which 10 are non-responders and 4 are responders. GSE93157 has 65 pre-treatment samples from melanoma ($n = 25$), lung non-squamous cancer ($n = 22$), lung squamous lung cancer ($n = 13$) and head and neck cancer ($n = 5$). Using the same response criteria, these samples are further grouped into non-responders ($n = 29$) and responders ($n = 36$). PRJEB23709 has 17 unique pairs of melanoma samples ($n = 34$) pre- and on-treatment, of which 7 are non-responders and 13 are responders. The UCLA cohort (PHS001919) has 60 unpaired pre- ($n = 27$) and post-treatment ($n = 33$) samples from melanoma. All these three data were mapped to the same gene space as CMAP2020, leading to the final dimension of $84 \times 10,157$ for GSE91061, for $14 \times 10,059$ for MGH cohort, $65 \times 766$ for GSE93157, $17 \times 9974$ for PRJEB23709, and $60 \times 10,059$ for UCLA cohort. Multiple biopsies from the same samples are grouped as one sample by taking the average during the analysis.

Gene expression and clinical data of TCGA patients were obtained from the GDC data portal (http://portal.gdc.cancer.gov). The analyses

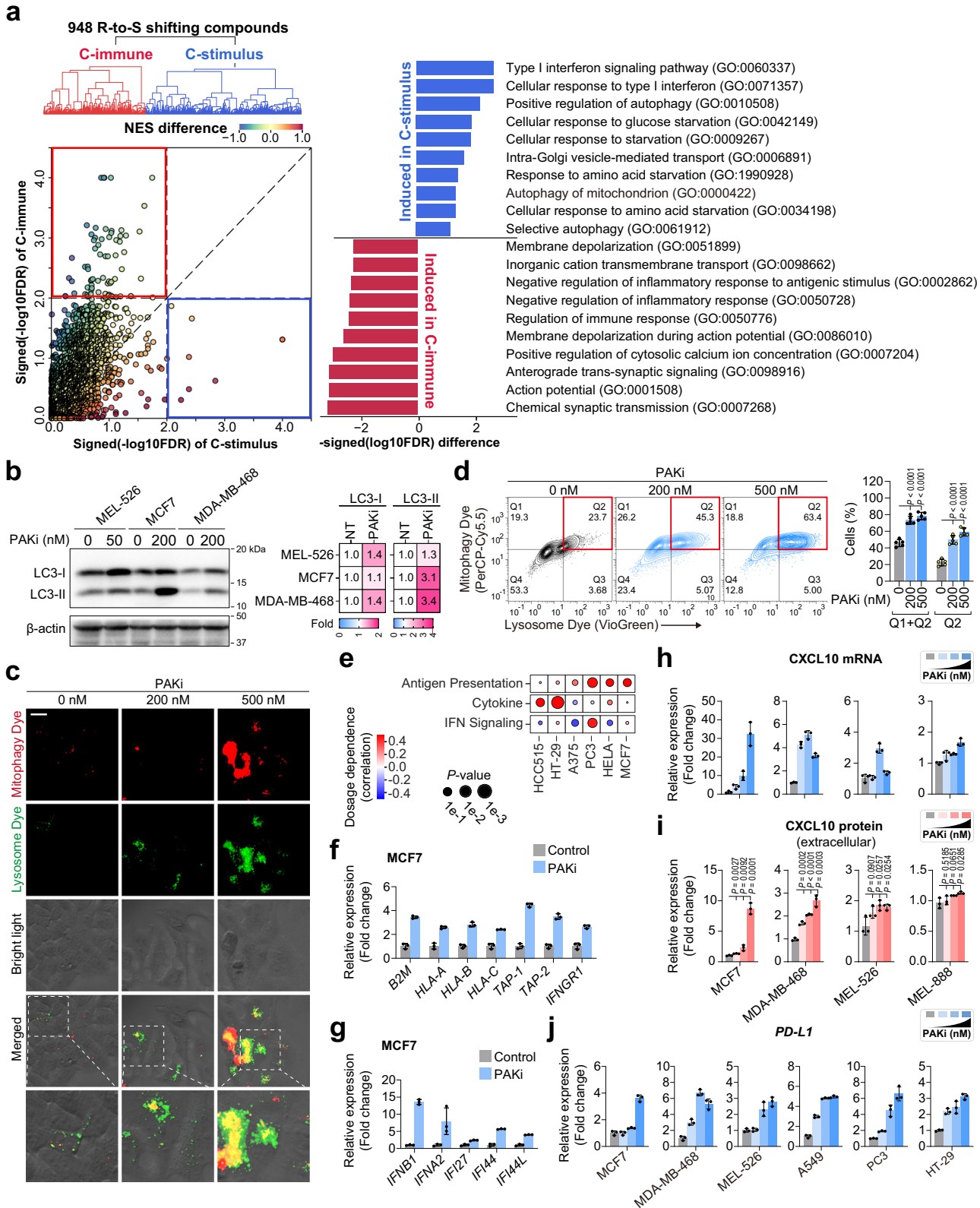

in this study were restricted to primary tumors except for melanoma where metastatic samples were focused, resulting in a total number of 10,004 bulk tumor samples. To evaluate the immunity contents, we used TIMER[65] to estimate the infiltration abundance of cytotoxic T cells, B cells, dendritic cells, macrophages, and nature killer cells in each bulk tumor sample.

## Classifying anti-PD-1 response using patient gene expression profiles

To evaluate the ability of using different gene expression profiles to classify anti-PD-1 response in patients, we applied PCA on treatment-naïve expression and treatment-induced expression of nivolumab-treated patients from GSE91061, respectively. For each patient,

**Fig. 5 | PAK inhibitor can induce mitophagy and immune response in cancer cells. a** Treatment induced expression analyses reveal mechanisms of chemo-immunotherapy synergisms. Source Data are provided as Supplementary Data 6. **b** Immunoblotting analysis of LC3 protein in cancer cells after 48 h of PF-03758309 (PAKi) treatment. The heat map (Right) indicates fold change of LC3-I and LC3-II band intensity, normalized to respective β-actin, DMSO served as a no-treatment control (NT). Experiments were repeated twice and obtained similar results. **c, d** PAKi treatment induces mitophagy in MCF7 cells. **c** Representative florescence microscopic images of MCF7 cells (24 h) labeled with mitophagy and lysosome dye, scale bar: 20 μm. Two independent experiments were performed and obtained similar results. **d** Flow cytometry detection of mitophagy in MCF7 cells, n = 5 biologically independent samples. **e** Pearson's correlation between

dosage and immunity signatures induction of PAK4 inhibitor PF-03758309 in multiple cancer cell lines. **f, g** qPCR validation of antigen presenting, processing genes (**f**), and interferon stimulated genes (**g**) in MCF7 cells after 48 h of PAKi treatment (200 nM). 0 nM or vehicle served as control, n = 3 technical replicates. Two independent experiments were performed and obtained similar results. **h, i** CXCL10 expression detected by qPCR (**h**) and ELISA (**i**) in cancer cells after 48 h of PAKi treatment. n = 3 technical replicates (**h**), biologically independent samples (**i**). **j** PAKi treatment induces *PD-L1* expression in cancer cells (48 h). n = 3 technical replicates. Concentration of PAKi used in (**h**–**j**): 0, 50, 200 and 500 nM for MCF7, MDA-MB-468, A549, PC3, and HT-29; 0, 2, 50 and 500 nM for MEL-526 cells. Data in (**d**) and (**f**–**j**) are presented as mean ± SD, P values in (**d**, **i**) were generated using a two-tailed Student's *t* test. Source data are provided as a Source Data file.

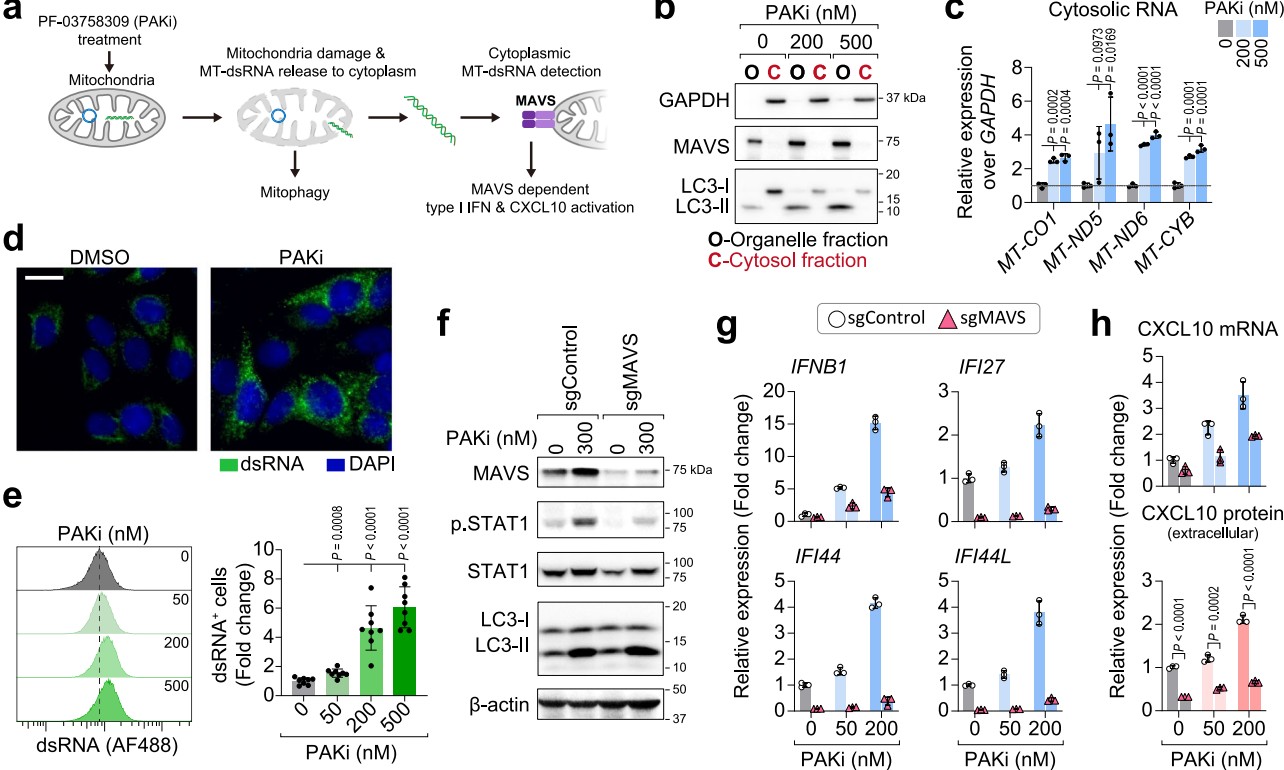

**Fig. 6 | PAK inhibitor-induced immune responses are mediated by mtRNA-dsRNA-MAVS. a** Schematic of PAKi induced mtRNA release and dsRNA-MAVS pathway. **b** Immunoblots express the purity of fractions from MCF7 cells treated with PAKi for 48 h. Cytosolic protein markers: GAPDH, LC3-I; organelle bound protein markers: MAVS, LC3-II. **c** qPCR analysis of mtRNAs in cytosol fractions from MCF7 cells treated with PAKi for 48 h. n = 3 biologically independent samples. **d** PAKi treatment induces dsRNA accumulation in MCF7 cells. Immunofluorescence analysis in 24 h DMSO or PF-03758309 treated MCF7 cells, Scale bar: 20 μm. **e** PAKi treatment induces dose-depended dsRNA expression in MCF7 cells (24 h). n = 8 biologically independent samples. **f** Immunoblotting analysis in MCF7 sgControl

and sgMAVS cells after 48 h of PAKi treatment. **g** qPCR analysis of *IFNB1* and interferon stimulated genes in MCF7 sgControl and sgMAVS cells after 48 h of PAKi treatment, n = 3 technical replicates. **h** CXCL10 expression detected by qPCR (top) and ELISA (bottom) in MCF7 sgControl and sgMAVS after 48 h of PAKi treatment. n = 3 technical replicates (top), biologically independent samples (bottom). Three (**b**, **c**) and two (**d**–**h**) independent experiments were performed and obtained similar results. Data in (**c**, **e**, **g**, **h**) are presented as mean ± SD, P values in (**c**, **e**, **h**) were generated using a two-tailed Student's *t* test. Source data are provided as a Source Data file.

we defined the response score as the first principal component of treatment-naïve/-induced expression profile, based on which patients are classified as non-responders and responders to nivolumab treatment. The classification performance was evaluated through receiver operating characteristic curve by comparing to the response groups defined in the original publication.

### Construction of anti-PD-1 response signature based on treatment-induced expression

To identify treatment-induced expression changes that robustly associate with anti-PD-1 response, we used GSE91061 as the main training set (n = 42), whereas PRJEB23709 (n = 17) and MGH cohort

(n = 14) are used as independent validation set (Supplementary Fig. 1b). For each paired sample in these three cohorts, the treatment-induced expression change of a gene is defined as the log2-transformed fold change between on-treatment and pre-treatment expression.

Within the major training set, we randomly chose 11 samples as leave-out validation set. The remaining 31 samples would undergo a training process with 3-fold cross-validation. Within each fold, samples will undergo 100 times of bootstrapping resampling with response-based stratification. Each bootstrapping will generate a resampled set $BS_{t,t\in[1,100]}$ of non-responders and responders. Differential expressed gene (DEG) analysis using student's *t* test will be applied to the resampled set. Genes with a *p* value less than 0.05 will be considered a

hit. Genes with higher expression in non-responders will be considered as a resistance (R) hit, whereas genes with higher expression in responders will be considered as a sensitivity (S) hit. After repeating this procedure 100 times, a hit frequency, named DEG selection score, will be generated for each gene.

$$DEGselection_{R(S)} = \sum_{t=1}^{100} hit_{R(S),t}/100, hit_{R(S),t} = \begin{cases} 0, p>0.05 \ in \ BS_t \\ 1, p<0.05 \ in \ BS_t \end{cases} \quad (1)$$

The candidate R signature $R_i$ and S signature $S_j$ will then be given by top $i$ percentage and top $j$ percentage of the DEG selection score on R hits and S hits. Using signature $R_i$ and $S_j$, an RS score $RS_{i,j}$ (difference between the enrichment of signature $R_i$ and $S_j$) will be calculated for each sample in the test set via GSEA pre-rank to evaluate if genes in $R_i$ or $S_j$ signature are enriched at the top or the bottom in ranked gene expression. The cross-validation performance of each pair of $i,j$ is given by the average AUC using $RS_{i,j}$ to classify patient response. The core-R or core-S signature will be then derived by overlapping the genes in $R_i$ or $S_j$ signatures from all three folds. We selected the final $i,j$ with an intention to keep a balanced number of genes involved in core signatures and to achieve a robust performance in cross-validation and leave-out validation set. The ability of final R and S signatures in recapitulating anti-PD-1 responses of patients from independent datasets are being validated in two independent validation cohorts, MGH and PRJEB23709.

We also validated our signatures in independent studies where sample-pairing is not available: GSE93157 and UCLA cohort. Since both of the datasets were not available for pre-on paired assessment, we used relative expressions differing from the cohort population baseline as a surrogate measurement of treatment-induced expression for each gene. The RS scores and the classification performance were obtained through the same aforementioned method.

Particularly, for genes in R signature, we observed that more than 74% of them are expressed in tumor cells, and approximately 50% of them also show expression in cancer-associated fibroblast, endothelial cells, as well as macrophages. For genes in S signature, we found quite some of them showed expression in macrophages, followed by T cells NK cells and tumor cells.

### Pathway and immunity assessment of R and S signature genes

To functionally annotate R and S signature, pathway enrichment analysis on cancer hallmarks is conducted by Gene Set Enrichment Analysis (GSEA). To demonstrate the immunity relevance of R and S signature genes in a broader range of cancers, we investigated how R and S signature genes can indicate immune cell infiltration in TCGA patient samples. Although TCGA patients did not receive immunotherapy, they have undergone intrinsic immune response processes which lead to the infiltration of immune cells into the tumor microenvironments. To this end, for patients from each cancer type, we used relative gene expressions differing from the population baseline as a surrogate measurement of expression changes. Average expression changes of genes from R/S signature were then compared to the TIMER estimation of immune cell fractions for each cancer type.

### Enrichment assessment of R and S signature and the calculation of shift ability score

To assess whether expression of R or S signature genes can be changed by a given perturbation $p$ ($p \epsilon$ {shRNA, compound}), we utilized the enrichment score calculation by pre-rank GSEA[66] with the weighting parameter set to 1. Specifically, for each perturbation $p$, a descending ranked gene list of size N, which contains treatment-induced expression changes of N genes {$g_1$, $g_2$, …, $g_N$}, is constructed according to CMAP level 5 signature. Normalized enrichment score of R signature and S signature will then be calculated through pre-rank GSEA and termed as $NES_R$ and $NES_S$, respectively.

To evaluate the potential a given perturbation $p$ ($p \epsilon$ {shRNA, compound}) can shift a cell line to an anti-PD-1 sensitive state, we created the concept of "shift ability". Briefly, the shift ability analysis will quantify the ability of a given perturbation in suppressing the R signature and inducing the S signature in a cell line. The shift ability score is thus given by the deviation from $NES_S$ of $NES_R$:

$$shiftability_p = \triangle NES = NES_S - NES_R \quad (2)$$

A high, positive shift ability means the perturbagen $p$ is able to suppress the R signature and at the same time promote the S signature, shifting the cell line to an immune-active and anti-PD-1 sensitive state. In contrast, a negative shift ability means perturbation will potentially cause immune suppression and anti-PD-1 resistance. Shift ability close to zero indicates the perturbagen might not be able to induce considerable shifting in immune response or have less effect on the immunotherapy efficacy.

### Immunity association of potent synergy targets

To evaluate the association between potent synergy targets and anti-tumor immunity in patients, we first collected 68 immune response gene signatures from previous studies[67]. An enrichment score was calculated for each signature using the single-sample gene set enrichment analysis[68] for each patient from TCGA cohorts. Enrichment scores of immune response signatures from the same immunity class would be averaged. Pearson's correlation was used to assess the association between patient immune response and potent synergy targets across different cancer types. An association with a $p$ value less than 0.05 would be considered as a significant correlation.

### Characterization of mechanism of chemo-immunotherapy synergism

Post-perturbation expression profiles (level 5 signature) of 948 R-to-S shifting compounds across different cell lines were extracted. For each compound, a consensus gene expression change signature will be calculated using the following method: for each gene across the samples from the experiments of the same drug, if its treatment-induced expression is higher than 1, the expression indicator will set as 1; if its treatment-induced expression is lower than −1, the expression indicator will set as −1; otherwise, the expression indicator will set as 0. For each compound, the sum of expression indicators across the samples will be used as its consensus expression change level. Pearson correlations were calculated and were utilized as distance metric between compounds in the gene space of 9196 best-inferred genes together with the landmark genes (10,174 genes in total). Based on the correlation matrix, hierarchical clustering analysis was performed using Ward method. Two major clusters were identified through the dendrogram cutoff at 12. The median value of the 10,174 genes' consensus expression changes will be ranked by descending and used as the consensus gene expression change signature of that corresponding cluster. For mechanism annotation of the major clusters, pre-rank GSEA was performed to assess the pathway enrichment in consensus gene expression change vectors using 3019 GO terms with 1000 times of permutation.

### General statistical analyses

For difference comparison, if not being particularly specified, Wilcoxon rank-sum test was applied to compare the differences between two unpaired groups; Wilcoxon signed rank test was applied to compare the differences between two paired groups; one-way ANOVA was applied to compare the differences between three or more groups; Kolmogorov–Smirnov test was applied to compare the differences between two continuous distributions. For correlation analysis, both Pearson and Spearman's correlation were applied in order to avoid the potential conflicts on linearity assumption. For enrichment and

exclusiveness, pre-rank GSEA was applied to assess the enrichment of specific features in single samples; hypergeometric test was applied for between-group comparison. For survival analysis, both Cox Proportional Hazard model and log rank test were utilized to compare prognosis between groups. All the computational and statistical analyses presented in this study were implemented by Python (version 3.8.0) in local or on the cluster of University of Pittsburgh Center for Research Computing.

## Animals

All mouse-related experiments were performed in full compliance with institutional guidelines and approved by the Animal Use and Care Administrative Advisory Committee at the University of Pittsburgh under Protocol #: 21099779. Female BALB/c mice, C57BL/6 mice, and FVB/NJ mice aged between 4 and 6 weeks were purchased from The Jackson Laboratories (CT, USA). Female mice were exclusively used because their more reliable behavior is expected to reduce overall variation in the data under various circumstances[69]. Mice were housed under pathogen-free conditions according to AAALAC (Association for Assessment and Accreditation of Laboratory Animal Care) guidelines. Mice were housed at an ambient temperature of 22 °C (22–24 °C) and humidity of 45%, with a 14/10 day/night cycle (on at 6:00, off at 20:00), and allowed access to food ad libitum.

In vivo antitumor efficacy was tested in syngeneic mouse colon (CT26) cancer models and mouse melanoma (B16) model. Female BALB/c mice or C57BL/6 mice were subcutaneously (s.c.) inoculated with CT26 cells ($5 \times 10^5$ cells per mouse) or B16 cells ($5 \times 10^5$ cells per mouse), respectively. When the tumor volume reached ~50 mm³, mice were randomly divided into four groups ($n = 5$) and treated with PBS (control), anti-PD-1 (5 mg/kg), Doxorubicin (2.5 mg/kg), and combination of anti-PD-1 (5 mg/kg) with Doxorubicin (2.5 mg/kg), respectively, every three days for a total of three times. Tumor sizes were monitored every three days following the initiation of the treatment and calculated by the formula: (Length × Width²)/2.

To evaluate the tumor infiltrated lymphocytes after treatment with anti-PD-1 and Doxorubicin, a syngeneic B16 melanoma model and MyC-CaP prostate cancer model were established by inoculating $5 \times 10^5$ B16 cells or $1 \times 10^6$ MyC-CaP into the flank of C57BL/6 mice or FVB/NJ mice, respectively. When the tumor volume reached ~50 mm³, mice were randomly grouped ($n = 5$), and treated with PBS (control), anti-PD-1 antibody (5 mg/kg), Doxorubicin (2.5 mg/kg), and combination of anti-PD-1 (5 mg/kg) with Doxorubicin (2.5 mg/kg), respectively, every three days for a total of three times. Tumor tissues were collected 1 day after the last treatment for further evaluation. The tumor growth in the study did not exceed the maximum size (2000 mm³) allowed by our institutional ethics committee.

## Cell lines and reagents

Human breast cancer cell lines MCF7 and MDA-MB-468, human melanoma cancer cell lines MEL-526 and MEL-888, human lung carcinoma cell line A549, human prostate cancer cell line PC3, and human colorectal adenocarcinoma cell line HT-29 were purchased from American Type Culture Collection (ATCC). MEL-526, MEL-888, and PC3 cells were cultured in RPMI-1640 (Hyclone). MCF7, MDA-MB-468, and A549 cells were cultured in Dulbecco's Modified Eagle Medium (Hyclone). HT-29 was cultured in McCoy's 5A Medium (Gibco). All cells were cultured with presence of 1X penicillin-streptomycin and supplemented with 10% heat-inactivated fetal bovine serum (Gibco) during normal growth conditions. Cells were cultured with antibiotic-free growth medium during drug treatment. Doxorubicin purchased from LC laboratories (#D-4000), anti-mouse PD-1 (#BE0033-2) and mouse IgG isotype control (#BE0086) from BioXCell. The PAK inhibitor PF-03758309 (PAKi) purchased from Selleckchem (#S7094). In nucleic acid depletion assay cells were treated with 10 μ/mL of DNase or 10 μ/mL of RNase (Invitrogen).

## Flow cytometry analysis of TIL

Flow cytometry was performed with LSRII (BD Biosciences) and Aurora (Cytek Biosciences) instruments and analyzed by FlowJo (BD Biosciences). B16 and MyC-CaP tumors were prepared for single cell suspensions. Briefly, tumors were dissected and transferred into RPMI-1640. Tumors were disrupted mechanically using scissors, digested with a mixture of deoxyribonuclease I (0.3 mg/ml, Sigma-Aldrich) and TL Liberase (0.25 mg/ml, Roche) in serum-free RPMI-1640 at 37 °C for 30 min, and dispersed through a 40 μm cell strainer (BD Biosciences). After red blood cell lysis, live/dead cell discrimination was performed using a Zombie NIR Fixable Viability Kit (BioLegend, dilution: 1/1000) at 4 °C for 30 min in PBS. Surface staining was performed at 4 °C for 30 min in FACS staining buffer (1× phosphate-buffered saline/5% FBS/0.5% sodium azide) containing designated antibody cocktails (PerCP anti-mouse CD45 antibody, Brilliant Violet 737 anti-mouse CD4 antibody, Brilliant Violet 615 anti-mouse PD-1 antibody, APC anti-mouse CD11b antibody, Brilliant Violet 510 anti-mouse Gr-1 antibody, APC/Cyanine7 anti-mouse F4/80 antibody, Pacific Blue anti-mouse MHC II antibody and PE anti-mouse CD163 antibody; dilution: 1/200 for all antibodies). For intracellular protein staining (FITC anti-mouse CD206 antibody; dilution: 1/200 for antibody), cells were fixed and permeabilized using the BD Cytofix/Cytoperm kit, following the manufacturer's instructions. For intracellular cytokine staining (PE-Cy7 anti-mouse IFN-γ antibody; dilution: 1/200 for antibody), cells were stimulated with phorbol 12-myristate-13-acetate (100 ng/mL) and ionomycin (500 ng/mL) for 6 h in the presence of Monensin. Cells were fixed/permeabilized using the BD Cytofix/Cytoperm kit before cell staining.

## Mitophagy assay

Mitophagy detection kit was purchased from Dojindo (Rockvile, MD) and used to detect mitophagy in PAKi and DXR treated cells according to the manufacturer's protocol. Cells were seeded on 96 well clear bottom black wall tissue culture plate. After 24 h, cells were washed twice with Hanks' Balanced Salt Solution (HBSS) followed by incubation with Mitophagy Dye at 37 °C for 30 min. After two washes cells, cells were incubated with or without drugs for 24 h. Cells then washed twice with HBSS and incubated with Lyso dye at 37 °C for 30 min. After two washes images were obtained using KEYENCE BZ-X800 fluorescence microscope.

For flow cytometry detection of mitophagy, cancer cells cultured in 6 well plates and washed two times with HBSS and incubated with Mitophagy dye at 37 °C for 30 min. After two washes cells, cells were incubated with or without drugs for 24 h. Then cells were collected by trypsinization, washed twice with HBSS and incubated with Lyso dye at 37 °C for 30 min. After two washes cells were suspended in HBSS and analyzed using MACSQuant analyzer (Miltenyi Biotec). Blue 655–730 nm (corresponds to PerCP-Cy5.5) and violet 525/50 nm (corresponds to VioGreen) fluorescence filters were used for Mitophagy dye and Lyso dye, respectively. Centrifugation steps were performed at $200 \times g$ in RT. Data were analyzed using FlowJo Software (10.9.0).

## Immunoblotting

Total proteins from specified cells extracted using RIPA lysis buffer contains 1X protease and phosphatase inhibitor. Protein concentration of each sample measured using BCA protein assay kit (ThermoFisher, #23227) as per the manufacturer's instructions. Then, equal concentration of proteins from each sample were taken and mixed appropriate volume of 5X protein sample buffer supplemented with reducing agent. Subsequent incubation at 98 °C for 10 min, equal amount of each protein sample was subjected to SDS-polyacrylamide gel electrophoresis using 4–12% gel and transferred to polyvinylidene difluoride membrane (Bio-Rad, #162-0177). The protein transferred membranes were immunoblotted with appropriate primary antibodies for overnight at 4 °C followed by appropriate horseradish peroxidase

(HRP) conjugated secondary antibodies at room temperature (RT) for 1 h. Signal was visualized by enhanced chemiluminescence substrate (ThermoFisher, #F32106) and exposed using iBright CL1500 imaging system (Invitrogen). Band intensities were quantified using ImageJ software and normalized to β-actin, heatmaps were generated using GraphPad Prism 9.5.1.

The following antibodies were used for immunoblotting analysis: rabbit anti-LC3A/B (CST, #12741, 1:1000), rabbit anti-STAT1 (CST, #14994, 1:1000), rabbit anti-phospho-STAT1 (CST, #7649, 1:1000), rabbit anti-STING (CST, #13647, 1:1000), rabbit anti-phospho-STING (CST, #50907, 1:1000), rabbit anti-MAVS (CST, #3993, 1:1000), mouse anti-GAPDH (Sigma-Aldrich, MAB374, 1:1000), and mouse anti-β-actin (Sigma-Aldrich, #A5441, 1:20,000). Goat anti-mouse and Goat anti-mouse IgG HRP conjugated secondary antibodies (SCBT, #SC-2005, #SC-2004).

### RNA isolation and quantitative real-time polymerase chain reaction (qPCR) assay

Total RNA from cells were isolated using TRIzol (ThermoFisher, #15596018) as per the manufacturer's instructions. Following RNA isolation, 1–2 μg of total RNAs used to synthesize cDNAs using High-Capacity cDNA Reverse Transcription Kit (Applied Biosystems, #4368813). qPCR analyses were performed using *Power* SYBR™ Green PCR Master Mix (Applied Biosystems, #4367659) in QuantStudio 6 Pro Real-Time PCR System (Applied Biosystems). Relative mRNA expressions were determined by calculating ΔΔCt values normalized to *GAPDH*, two-tailed Student's *t* test used to calculate statistical values. Sequences of primers used for qPCR are given in Supplementary Table 5.

### Enzyme-linked immunosorbent assay (ELISA)

Cell culture supernatants after treatment were used for the quantification of cancer cells secreted CXCL10 using ELISA. The human CXCL10/IP-10 ELISA kit was purchased from R&D Systems (#DY266). After the removal of cell debris by centrifugation, ELISA was performed according to the manufacturer's protocol. The optical density was determined using a Bio-Rad xMark™ Microplate Absorbance Spectrophotometer.

### Cloning, sgRNA construction and lentiviral transduction

We designed three guide RNA (gRNA) for MAVS and one scrambled gRNA as a control and used lentiCRISPRv2 vector. Lentiviral particles were prepared after transfection of plasmids into HEK-293T cells using Lipofectamine 2000™ (Invitrogen, #11668019). Targeted cells were infected with the lentivirus packaged by Cas9 and single-guide RNA (sgRNA) expression plasmid encoding puromycin resistance (Addgene plasmid, #52961). The knockout efficiency of MAVS was determined by the immunoblotting and qPCR analysis after selection of puromycin resistance cells. gMAVS_F3/R3 exhibited the maximum knockout efficiency of MAVS gene in MCF7 cells and were used for subsequent experiments. Guide RNA sequences used to generate MAVS knockout:

gMAVS_F1: caccgCTTCCGGTCGGCTTGTGGCC;
gMAVS_R1: aaacGGCCACAAGCCGACCGGAAGc;
gMAVS_F2: caccgAGGTGGCCCGCAGTCGATCC;
gMAVS_R2: aaacGGATCGACTGCGGGCCACCTc;
gMAVS_F3: caccgGTGTCTTCCAGGATCGACTG;
gMAVS_R3: aaacCAGTCGATCCTGGAAGACACc

### Immunofluorescence analysis of dsRNA

For immunocytochemical analysis, the cells were cultured in 8-well chambered slides (Thermo Scientific, #154534) and treated as previously described. After three washes with PBS, cells were fixed with 4% formaldehyde for 20 min at RT. After two washes cells were incubated with a blocking-permeabilization buffer (5% goat serum and 0.3% Triton X-100 in PBS) for 1 h. Then cells were incubated with anti-dsRNA (J2) antibody diluted (2.5 μg/mL) in antibody diluent buffer (1% BSA in PBS) overnight at 4 °C in a humidified chamber. After three 5 min washes with PBS, the cells were incubated with Alexa Fluor 488 conjugated goat anti-mouse IgG H&L antibody (2 μg/mL) (abcam, #ab150113) at RT for 1 h in the dark. After three 5 min washes, the cells were mounted using the ProLong™ Gold Antifade Mountant with DAPI (Invitrogen, #P36935) and coverslips. Slides were imaged using a KEYENCE BZ-X800 fluorescence microscope.

### Flow cytometry analysis of dsRNA

After 24 h treatment, cells were collected by trypsinization and washed twice with PBS. Cells were fixed with 4% formaldehyde for 20 min at RT. After two washes with PBS, cells were permeabilized for 15 min at RT using 0.1% Triton X-100 in PBS. Cells then incubated with 1% BSA for at RT for 1 h followed by incubation with 2.5 μg/mL of anti-dsRNA (J2) antibody (CST, #76651) at RT for 1 h. After three washes cells were incubated with 2.2 μg/mL of Alexa Fluor 488 (AF488) conjugated goat anti-mouse IgG H&L antibody (abcam, #ab150113) at RT for 1 h. Cells then washed three times with PBS and suspended in 0.5% BSA, 2 mM EDTA in PBS. Centrifugation steps were performed at $300 \times g$ in 4 °C. Cells were analyzed using MACSQuant analyzer (Miltenyi Biotec). Data were analyzed using FlowJo Software (10.9.0).

### Cell fractionalization and mtRNA detection

Cytosolic and organelle fraction were prepared as previously described[70]. 25 μg/mL of digitonin (Millipore, #300410) was used to isolate cytosolic and organelle fractions. Following isolation of the cytosolic fractions, the remaining crude fractions were washed 3 times using PBS and served as organelle fraction containing the nucleus, mitochondria, etc. Putity of fractions were analyzed using immunoblotting and qPCR analysis. Total RNA from cytosolic and organelle extracts after treatment were isolated using TRIzol. Following cDNA synthesis, qPCR analyses were performed with mitochondria gene specific primers as previously described[54] and normalized to *GAPDH*.

### Reporting summary

Further information on research design is available in the Nature Portfolio Reporting Summary linked to this article.

## Data availability

The shift ability score and the experiment data generated in this study are present in the paper and/or Supplementary Information. The publicly available post-treatment cell line transcriptome data used in this study, including shRNA and compound treatment, are available at the Expanded Connectivity Map (CMAP) LINCS Resource 2020 (complete version, 11/23/2021) through the CLUE portal (http://clue.io). The publicly available gene expression and clinical data of melanoma patients used in this study are available in the Gene Expression Omnibus (GEO) with accession number GSE91061, GSE168204, GSE115821 and GSE93157 and in the European Nucleotide Archive (ENA) with accession number PRJEB23709. The publicly available gene expression and clinical data of The Cancer Genome Atlas (TCGA) patients can be obtained from the GDC data portal [http://portal.gdc.cancer.gov]. All the remaining data are available within the Article, Supplementary Information or Source Data file. Source data are provided with this paper.

## Code availability

The code for anti-PD-1 response signature construction and shift ability calculation in this study has been deposited at https://github.com/DaYangLab2015/ChemoImmunoSyng[71].

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

## Acknowledgements

We highly appreciate Drs. Antoni Ribas and Gabriel Abril Rodriguez for kindly sharing with us the patient and in vivo data. We also thank the Center for Simulation and Modeling (SaM) at the University of Pittsburgh for computing assistance. This work is supported by The Shear Family Foundation (to D.Y.), The American Cancer Society Research Scholar Award (132632-RSG-18-179-01-RMC) (to D.Y.), and The National Cancer Institute (R01CA222274, R01CA255196, R01CA272866, and R01CA282704) (to D.Y.)

## Author contributions

Conceptualization: Y.W., M.Z., D.Y., Methodology: Y.W., D.P., Investigation: Y.W., D.P., H.H., Y.Z., S.H.L., Y.F.W., Visualization: Y.W., D.P., H.H., Funding acquisition: M.Z., D.Y., Project administration: Y.W., M.Z., D.Y., Supervision: M.Z., S.L., D.Y., Writing – original draft: Y.W., D.P., H.H., Y.Z., M.Z., D.Y., Writing – review & editing: Y.W., D.P., H.H., Y.Z., S.H.L., Y.F.W., M.Z., D.Y.

## Competing interests

The authors declare no competing interests.
