## [Peer Review File · Nature Communications]

Systematic investigation of chemo-immunotherapy synergism to shift anti-PD-1 resistance in cancerREVIEWER COMMENTS

Reviewer #1 (Remarks to the Author): with expertise in computational biology, cancer, immunotherapy

The topic of predicting chemo-immunotherapy synergisms that help boost cancer ICB treatment is highly interesting and important. The use of a computational approach leveraging the transcriptomic data is a potentially effective approach to facilitate this goal. The paper is well-written, the logic is clear, and the validation using mouse model is useful, however, not very strong. Following are a few concerns for the authors to consider to further improve the manuscript.

Major:

1. The R and S signatures are derived from differential gene expression analysis related to immunotherapy response. Interestingly, these signatures also appear to show a strong correlation with survival in TCGA. Therefore, a question arises as to whether these signatures are predictive or simply prognostic.
2. It is crucial to perform a careful check of the code output and thoroughly proofread the manuscript. Currently, there are some inconsistencies throughout the paper. For instance, in line 71, it mentions "68 melanoma patients," but in SFig. 1, there are 41 patients, and in the Methods, it indicates that there are 42 patients. In another example, in line 79, it mentions 130 S genes, but in the Methods, there are 139 genes.
3. Regarding lines 344-345: In dataset GSE91061, the SD group was grouped with PR/CR as responders, which was further used to derive R and S signature genes. While this grouping may have been used in the original paper, it's more common to group PD/SD as non-responders and CR/PR as responders. Could the authors consider using this grouping method or excluding the SD group from the analysis to determine if the results remain robust?
4. In reference to line 81, the author stated the utilization of three cohorts for evaluating the predictive capabilities of predicting to anti-PD-1 response with the R and S signature. However, Figure 1d displays a total of four cohorts, excluding the training dataset. It would be helpful to provide clarification regarding the specific cancer types that were tested and whether the treatment conditions were pre-treatment or post-treatment.
5. Generally, there is a larger number of publicly available cohorts treated with immune

checkpoint blockade (ICB). However, this manuscript has only utilized a limited subset of these cohorts. We encourage the authors to consider incorporating additional datasets to strengthen the robustness of their findings.

6. In the final paragraph of the first results section, the authors examine the combination of S and R signatures; however, they do not provide a clear explanation of the methodology behind this combination. Furthermore, when compared to S and R individually, does the combination of these two signatures demonstrate superior performance?

7. In the manuscript, various measurements of the S and R signatures were employed, including PCA, average expression of signatures, and Gene Set Enrichment Analysis (GSEA). It is advisable to maintain consistency in the measurement method throughout all analyses. Alternatively, if different measurement methods are used, the authors should offer a rationale for their choices.

8. In Fig. 5d, this represents the only experimental validation in the study. Is there a statistically significant difference between the anti-PD-1 vs. anti-PD-1+DOX and between the DOX vs. anti-PD-1+DOX groups? If not, the validation may not be considered adequate, particularly from a statistical perspective.

Minor:

9. In lines 74-75, please clarify what the prediction score for each patient used to calculate the AUC is.

10. Lines 77-79: How are the numbers of S genes and R genes determined? Please provide clarification on this.

11. In Fig. 1a, d, SFig. 1e, g, and relevant main text, it's important to specify the sample size (and may also the p values and 95% CIs) for each AUC calculation.

12. In SFig. 1b and c, the color code appears suboptimal as it can be challenging to distinguish SD from PR/CR. Consider improving the color differentiation.

13. In SFig. 1e, could the authors also compare AUCs between pre- and post-treatment in the original set for additional context?

14. In SFig. 2a, please display the correlation coefficient and p-value on the figure.

15. It would be helpful to add the numbers at risk at the bottom of the Kaplan-Meier curves.

16. Check the consistency between the text (lines 108-111) and the figure panels.

17. In Fig. 2g and h, ensure that the legend and figure are consistent.

18. It would be beneficial to include 95% confidence intervals for the hazard ratios (HRs).

19. At line 80, please ensure that the citation for the figure is accurate.

Reviewer #2 (Remarks to the Author): with expertise in bioinformatics and systems genomics

In this study, Wang and colleagues studied the gene expression difference between immunotherapy responders and non-responders of melanoma patients and identified signatures expressed higher in responders (S signature) and reversely higher in the latter (R signature), and then crossed the signature genes with cancer cell line transcriptomics data treated with chemical compounds or shRNAs. The results validated the immune response relevance and clinical outcome relevance of the signature genes and identified compounds that may enhance anti-PD-1 immune therapy. Very importantly, the authors further performed cell line and mouse study to validate the strong candidates and to elucidate the potential mechanism. The study approaches and results are both very interesting and comprehensive, with strong bioinformatics analysis and experimental validation. The manuscript is overall very well written and presented clearly, except a few places. The work could be significant to the field for developing combined treatment. I don't have much more to contribute to improve the manuscript significantly, because it is already excellent. Here are a few comments that the authors may evaluate if they are helpful.

1. It would be interesting to know how many of the R/S signature genes are expressed in different TME cell types highly or specifically, particularly immune cells, like macrophages and NK cells. This is relevant, because the observed gene expression difference between patients may reflect the cell composition change in their TME.
2. In related to 1, as transcriptomics from different platforms were used in the studied, some including all human genes other not. Can the 1190 R and 130 S genes be further reduced to smaller core sets that are consistently present across data and remain predictive?
3. The expression for most genes in the CMAP was derived from imputation, inferred from their expression correlation with the landmark genes that were experimentally measured. Is this a concern in the usage of the data for computing scores for R/S genes sets, as genes in the same set are likely correlated?

4. In a few places, the text may be expanded a little to include more details. This can be achieved and will be helpful to readers. The current journal doesn't have word limitation. For examples, the first line in Result, "We obtained the paired transcriptomic data ..." The authors can simply clarify if the data is from blood or tumors, even though the information is in M&M. Following this, "PCA revealed that the anti-PD-1 induced expression achieved a better classification performance (AUC=0.77)..." This statement seems to condense multiple analyses into one; PCA analysis of what? How does PCA connect to AUC? Your readers are not necessary all computation researchers.

5. The authors found that post-treatment transcriptomics data are better for prediction than pre-treatment ones. So, how many of the R/S genes showed differential expression before and after anti-PD-1 treatment? As the goal for clinical usage, we probably want to consider pre-treatment data. Most of the other gene expression data analyzed in the manuscript are not from post anti-PD-1 treatment either. Would excluding the differentially expressed genes before and after treatment from the R/S signature genes have any impacts on the results and key conclusions? In related to this point and #3 above, please clearly indicate if data is from patients before or after treatment in the text when patient data are described in each analysis.

6. In evaluating the prediction performance, do some patient (or cell line) samples contain replicates? How much variation exists for the results between replicate samples, e.g., R-to-S shift?

Reviewer #3 (Remarks to the Author): with expertise in cancer immunotherapy

In this study, Dr. Yang and colleagues analyzed a number of public datasets to identify the gene expression signature that may predict tumor sensitivity to anti-PD1 therapy, and, further, they showed that perturbation of resistant genes by chemotherapy can cause a shift to the sensitive gene signature. Mitophagy pathway is proposed as a potential target for the combination of chemotherapy and immunotherapy. However, there is a lack of sufficient experimental evidence to support the strong claim of chemo-immunotherapy synergism.

1. In Fig. 1, the author begun with a computational analysis of paired transcriptomic data

from melanoma specimens, but then jumped to the analysis of multiple types of cancers, in which the transcriptomic data are probably not paired. The connection is not quite clear to me. Why in Fig. 1a PR and CR are combined for the analysis, while they are separated in Fig. 1c? Average gene expression doesn't show a pervasive difference between PR and PD, in terms of either R signature or S signature.

2. The inverse correlation between T cell infiltration and R signature is not strong in cancer types other than SKCM in Fig. 2d. It raises a question whether the finding should be only applied to certain cancer types, such as melanoma, rather any cancer types.

3. Different types of cancers have diverse transcriptomics and distinct tumor microenvironment, so it is surprising to me that perturbation of a single given R gene or S gene can cause a strong signature shift in many cancer types.

4. When intersecting the knockdown analysis with drug treatment analysis, with regard to shift ability, how is the correlation or reproducibility between these two kinds of treatments?

5. The data from CT26 model in this study does not support a strong claim of synergism, because either anti-PD1 or dox alone already inhibits tumor growth a lot. The TIL analysis should be performed with four groups of treatments as in Fig. 5d to show the immunological basis of synergism.

6. In Fig. 6, two new cell lines were introduced for investigation of mitophagy. Why the cell lines in the previous figures were not used here to make data consistent and comparable?

7. There seems to be a technical issue with the immunoblots of phospho-STING in Fig. S6f and g.

8. Mitophagy removes the damaged or dysfunctional mitochondria, which is a key source of intracellular dsRNAs. How could the enhanced mitophagy by PAK inhibition generate more dsRNAs? Are other pathways also involved when targeting PAK? Does PAK inhibitors

sensitize refractory tumors to anti-PD1 in syngeneic tumor models? In vivo immunological analysis should also be performed.

Reviewer #4 (Remarks to the Author): with expertise in computational biology, oncology

The authors compare gene expression of paired pre- and on-treatment patient transcriptomics data to identify gene signatures for sensitivity and resistance to anti PD-1 therapy. Using these signatures, they introduce a shift ability score to assess immune response of chemo-/targeted therapy and link it to the potential synergistic effect of anti PD-1 therapy. They perform multiple in silico and in vivo experiments to show the validity of their approach. They characterised biological mechanisms that potentially can increase the efficacy of PD-1 immunotherapy treatment when used in combination with chemo-/targeted therapy. Identification and experimental validation of mitophagy as immunotherapy and chemo-/targeted therapy synergism mechanism is an important finding that further investigated and translated to clinical practice can benefit patients. However, the manuscript has several shortcomings, please see below.

Comments

Authors write about chemo-immunotherapy combinations and synergies, however the term “immunotherapy” in the manuscript is limited to the single PD-1 inhibitor (nivolumab) and the term “chemotherapy” seems to include kinase inhibitors (e.g. MEK, RAF inhibitors). I suggest treating both terms more precisely.

The Introduction section gives a very high-level overview of the combinatorial cancer treatment and interaction between immunotherapy and chemo-/targeted therapy accessible to the general audience, whilst graces literature related to the main section. For example, no mention of the PD-1 function and inhibitor (a single immunotherapy considered in analysis) is available. Desirable would be a structured overview of related work in terms of application and methodological domain.

The R and S signatures were determined based on the 43 patients only. Do the authors

consider their approach to be statistically robust? For example, bootstrapping or cross-validation may assess this. Including 1,329 genes in R and S signatures may also include many false positives and noise. Also, a power analysis may be beneficial.

What is the main reason for using TCGA data for validation? It does not assist in anti PD-1 therapy and synergistic effect evaluation. Also, the authors write: "Although TCGA patients did not receive immunotherapy, they have undergone intrinsic immune response processes which lead to the infiltration of immune cells into the tumor microenvironments." It is not clear why they set it similar to the anti PD-1 therapy. I assume that also in immunotherapy, intrinsic immune responses take place. To reduce the results overload excluding analysis on the TCGA data or replacing it using data from Gene Expression Omnibus (GEO) can be considered.

The authors write that "combining 181 the compound and shRNA screening result revealed 49 drug targets whose genetic and 182 pharmacological inhibition can induce R-to-S shifting in the same cell lines." They provide some examples of those targets, e.g. BRAF 28, RRM1 10,11, CDK129, CDK430, HDAC123, TOP1 as well as PAK4 (described as novel target). They further experimentally validated in vivo the synergistic effect of anti PD-1 therapy and doxorubicin. However, this effect has been already described in the literature. What is the rationale behind the choice of target to validate experimentally? And are there any novel, previously undescribed drug targets discovered?

The authors use three different datasets for their analysis. Why did they decide not to harmonize the transcriptomic data coming from different sources (e.g. batch correction, tissue correction). However, if they did, it would be great to provide the details on the used approach.

The interpretation of the introduced shift ability score (line 421) is not completely clear since the enrichment score (for both ES_R and ES_S) covers genes that are over-represented at either the top or bottom of the ranked list. Could you please elaborate more on the interpretation of shift ability score (lines 422-427)?

The code repository is very unstructured. It further does not contain any instructions on how to set up a working environment and run the code. With the initial code state, the analysis results can be barely reproduced.

The performed analysis workflow is separated into different subsections (e.g. Predicting anti-PD-1 response using patient gene expression profiles or Construction of anti-PD-1 response signature based on treatment-induced expression, etc). However, it still remains very unstructured, the steps when and after the shift scores are calculated are not easy to understand. I suggest introducing more structure into the methods (and consequently results section) including clearly indicating the main goal of this analysis parts.

Also, the current manuscript contains a lot of results which are not very critical for the main contribution in identifying potential synergistic targets. For instance,

Classification in Treatment-induced expression change profiling can predict anti-PD-1 response in patients subsection (lines 71-76)

All validation experiments on GSE93157, GSE168204 and some experiments on theTCGA dataset (lines 80-91, and more)

Survival analysis (lines 112-118)

Etc

Certainly, the experiments are of high importance for justification and validation of every analysis step, they impede the understanding of the main manuscript message. The authors can consider moving the external validation analysis and corresponding figures in the supplementary materials. This would improve the readability and understanding of the manuscript significantly.

The approach to calculate R and S signatures and use it for novel target discovery can be potentially interesting for other applications. What do authors think about it? Or is there anything that limits its applicability only to PD-1 combination therapy? What could be the follow-up or future work? Please, address it in the discussion section.

Minor comments

In the introduction section, no references are provided for multiple statements making the writing sound as an opinion. Below are only examples, I recommend the authors to revise their citations:

-) “Even for the patients who initially respond to the therapy, the later developed immunotherapy resistance remains to be challenging” (lines 37-38)
-) “However, the design of the combination regimens so far is largely relied on clinical experiences, it is very challenging to characterize new chemo-immunotherapy synergisms. The emerging large-scale pharmacological transcriptomic datasets that profile the expression changes after drug/immunotherapy treatment provide us deeper and novel insights on how treatment changes biological processes in the tumor.” (lines 52-55)
-) “Of note, by 2023, FDA have approved several chemo-immunotherapy regimens in diffuse large B-cell lymphoma (Polatuzumab + bendamustine/rituximab), triple-negative breast cancer (Atezolizumab/Pembrolizumab + taxanes), gastric cancer and esophageal adenocarcinoma (Nivolumab + FU-/platinum).” (lines 44-48)

Why first infer genes & increase dimension to reduce it in the analysis later again? (Lines 33-339) How do the authors define best inferred genes? How is the inference quality assessed?

The authors use different numbers of melanoma patients in the GSE91061 dataset used to determine S- and R-signatures. Please consider updating the inconsistent numbers:

-) 42 patients used for the analysis (lines 343-347)
-) 68 patients in the results section (line 71)
-) 41 patients in Figure S1a

Authors use “gene expression” as synonym to “gene expression changes” in multiple occurrences, when describing the “log₂-transformed fold change between on- treatment and pre-treatment expression”. For example,

-) “ In 9,626 patients across 23 cancer types from TCGA database, we observed a dramatically negative correlation between expression of R genes and S genes...”
-) Specifically, we observed that S signature expression are strongly correlated with immune-hot phenotypes...” (line 96)

-) "Increased expression of R signature is highly correlated with decreased infiltration of CD8+ T cells..." (line 98)

-) "Y-axis represents the Wilcoxon rank-sum test of gene expression on treatment-induced level between anti-PD-1 sensitive patients and resistant patients." (lines 745 - 747, Figure 1b)

It is not clear what PHS001919 (line 82) refers to. It is not described in the materials and methods section.

Some terms are not defined before the first usage. For example, the authors do not explicitly explain how they map PD, ST, PRCR response scores (lines 342-354) to responders and non-responders (368-369, results section and figures)

In the result section, the authors may have incorrect references to the subfigures of Figure 1:

-) "Fig. 1a, b..." line 75 should be Fig. 1a

-) "Fig. 1c" line 80 should be Fig. 1b

Figure. It is not clear what "in different timepoints" in the sentence "ROC curve shows the classification performance in different time points." refers to. (lines 772-773, Figure S1e)

There might be a contradiction in the last paragraph of the discussion section. What is their conclusion?

-) "The mitophagy-mediated release of mtDNA and mtRNAs activates the anti-viral signaling, which will initiate the innate immune response and induce the CXCL10 secretion. Both innate immune response and CXCL10 secretion have been demonstrated to increase the efficacy of immune checkpoint blockades"

-) "Future clinical studies are warranted to determine whether drug-induced mitophagy should be minimized to control chemo-resistance or be exploited to synergize immunotherapy."

The authors denote classification models with AUC between 0.66 and 0.75 as "robust".

(Lines 87-88). In my opinion, robust models are defined differently and are comprehensively tested.

The paper concludes that “Future clinical studies are warranted to determine whether drug-induced mitophagy should be minimized to control chemo-resistance or be exploited to synergize immunotherapy.” It is not clear why authors refer directly to clinical trials and skip in vitro and in vivo validation of the drug-induced mitophagy for immunotherapy and chemo-/targeted therapy synergism.

Summary: We sincerely appreciate the constructive comments from all the reviewers. In this revised manuscript, we have addressed all of the concerns of the reviewers and summarized our major changes below followed by a detailed point-by-point response.

1. We have refined our signature construction and evaluation procedures for anti-PD-1 response by **(i)** incorporating the bootstrapping resampling and cross-validation; **(ii)** validating the signature performance in two additional patient cohorts with paired pre- and after-anti-PD-1 treatment samples, and **(iii)** investigating if the new signature is derived from the changes of the TMB using scRNA-seq data from before- and after-treatment. The new signatures now have balanced gene numbers and better performance.
2. Based on the refined signatures, we have redone all the downstream analyses including synergistic prediction and mechanism investigation. We have optimized the method of calculating the shift ability score. The resulted number of potential targets reduced from 17 (first submission) to 14.
3. For the validation of synergistic effects between doxorubicin and anti-PD-1 therapy, we have redesigned and redone the experiments in original syngeneic mouse models (e.g., CT26) and added prostate cancer (e.g., MyC-CaP) and melanoma syngeneic mouse models (e.g., B16). We have performed immune profiling of tumors based on the reviewer's suggestions.
4. Mechanistically, we have now included data from three additional cell lines besides MCF7, MEL-888, and MEL-526 to validate our prediction that PAKi treatment activates type I interferon pathway. We have performed the mitophagy detection assay and demonstrated that both doxorubicin and PAKi treatments activate mitophagy in a dosage dependent manner. We have also specifically detected mitochondrial mRNA expression in organelle and cytosol to demonstrate that treatment-induced cytosolic mtRNA activates type I interferon pathway.
5. We have included a discussion of the limitations and the prospective application of our approach. We have also updated the introduction section with a particular focus on anti-PD-1 therapy.

Detailed Point-by-point response to REVIEWER COMMENTS

Reviewer #1 (Remarks to the Author): with expertise in computational biology, cancer, immunotherapy

The topic of predicting chemo-immunotherapy synergisms that help boost cancer ICB treatment is highly interesting and important. The use of a computational approach leveraging the transcriptomic data is a potentially effective approach to facilitate this goal. The paper is well-written, the logic is clear, and the validation using mouse model is useful, however, not very strong. Following are a few concerns for the authors to consider to further improve the manuscript.

Major:

1. The R and S signatures are derived from differential gene expression analysis related to immunotherapy response. Interestingly, these signatures also appear to show a strong correlation with survival in TCGA. Therefore, a question arises as to whether these signatures are predictive or simply prognostic.

We thank the reviewer for this insightful comment. We agree that TCGA data is not a rigorous validation of our signature genes and have moved the majority of the TCGA data into supplementary. In the revised manuscript, we have (1) redone the analysis and identified a more compact set of signatures and (2) included additional before-after treatment paired transcriptomic

data as independent validation (**lines 81-84, Fig. 1, Extended Data Fig. 1, and method lines 372-412**). The results show that our refined signatures have balanced gene numbers and robust classification performance. We have also performed experimental validation in additional mouse and cell lines models to demonstrate the predicted synergistic effects. In the discussion section, we have now clarified that the best application of our study is not to predict individual patient response. Instead, the signature identified in our study served as a tool to find promising drugs that are synergistic with anti-PD-1 therapy.

2. It is crucial to perform a careful check of the code output and thoroughly proofread the manuscript. Currently, there are some inconsistencies throughout the paper. For instance, in line 71, it mentions "68 melanoma patients," but in S Fig. 1, there are 41 patients, and in the Methods, it indicates that there are 42 patients. In another example, in line 79, it mentions 130 S genes, but in the Methods, there are 139 genes.

We thank the reviewer for pointing out these details. We have carefully proofread the revised manuscript and corrected the numbers.

3. Regarding lines 344-345: In dataset GSE91061, the SD group was grouped with PR/CR as responders, which was further used to derive R and S signature genes. While this grouping may have been used in the original paper, it's more common to group PD/SD as non-responders and CR/PR as responders. Could the authors consider using this grouping method or excluding the SD group from the analysis to determine if the results remain robust?

We had the same question that SD should be categorized as non-responder based on our previous experience on chemotherapy. After consulting with our clinical collaborators, we learned that immunotherapy is slightly different from conventional chemotherapy and targeted therapy because of a phenomenon called "pseudo-progression". Based on the iRECIST guideline that specifically designed for immunotherapy response evaluation in clinical¹, patients with stable disease (i.e., SD) is usually considered as "beneficial". This is probably the reason in the original paper, the SD patients were grouped as responders.

To further address the reviewer's concern, we also conducted analyses that exclude the SD patients, or group the SD with PD patients. Interestingly, we identified that most of the immune response happened in SD patients instead of PR/CR patients, indicating SD is an intermediate stage between PD and PR/CR (See the below figure). With the above guideline information and this observation, we decide to align with the original paper, which groups the SD patients and PR/CR patients together as "pharmacological responders".

Pathways enriched in PRCR patients

Pathways enriched in SD patients

Pathways enriched in PD patients

Figure legend: Top pathway enrichment in genes highly expressed in PRCR (upper), SD (median), PD (lower) patients. X axis represents negative log₁₀-transformed adjusted P-value from GSEA enrichment analysis.

4. In reference to line 81, the author stated the utilization of three cohorts for evaluating the predictive capabilities of predicting to anti-PD-1 response with the R and S signature. However, Figure 1d displays a total of four cohorts, excluding the training dataset. It would be helpful to provide clarification regarding the specific cancer types that were tested and whether the treatment conditions were pre-treatment or post-treatment.

In the revised manuscript, we have added the treatment conditions and cancer type information to the updated figures (Extended Data Fig. 1k).

5. Generally, there is a larger number of publicly available cohorts treated with immune checkpoint blockade (ICB). However, this manuscript has only utilized a limited subset of these cohorts. We

encourage the authors to consider incorporating additional datasets to strengthen the robustness of their findings.

In the revised manuscript, we have now performed a thorough study and find two independent cohorts that have transcriptomic data from pre- and post- treated tumor samples. We have included those two independent cohorts (MGH cohort and PRJEB23709) in the validation procedure. The related results can be found at **lines 81-94, Fig. 1c, d, e, and method lines 413-417.**

6. In the final paragraph of the first results section, the authors examine the combination of S and R signatures; however, they do not provide a clear explanation of the methodology behind this combination. Furthermore, when compared to S and R individually, does the combination of these two signatures demonstrate superior performance?

We have compared the classification performance using R or S signature separately. In the leave-out validation set, using R signature alone (AUC=1.0) did show better performance than using S signature (AUC=0.63) or the combined signature (AUC=0.8). However, in the independent validation sets, the combined signature showed a more robust performance (AUC=0.74 for PRJEB23709 cohort, 0.7 for MGH cohort) than using R signature (AUC=0.68 for PRJEB23709 cohort, 0.56 for MGH cohort) or S signature (AUC=0.71 for PRJEB23709 cohort, 0.64 for MGH cohort) alone. It appears that the combination of the R and S helps overcome the overfitting in training. We thank the reviewer for guiding us to perform this analysis. We have also added detailed description in the method section at **lines 397-407.**

7. In the manuscript, various measurements of the S and R signatures were employed, including PCA, average expression of signatures, and Gene Set Enrichment Analysis (GSEA). It is advisable to maintain consistency in the measurement method throughout all analyses. Alternatively, if different measurement methods are used, the authors should offer a rationale for their choices.

In the revised manuscript, we have now updated our methods to GSEA pre-rank for most of the cases when calculating the signature scores. We have included the measurements information in the method section.

8. In Fig. 5d, this represents the only experimental validation in the study. Is there a statistically significant difference between the anti-PD-1 vs. anti-PD-1+DOX and between the DOX vs. anti-PD-1+DOX groups? If not, the validation may not be considered adequate, particularly from a statistical perspective.

In the revised manuscript, we have redone the mouse model analysis by including more mice in each treatment group for the CT26 model. To further address the reviewer's concern, we have tested the anti-PD-1+DOX combination in one additional B16 murine melanoma model. The statistical significance has been included in both mouse models.

Minor:

9. In lines 74-75, please clarify what the prediction score for each patient used to calculate the AUC is.

In the revised manuscript, we have included a detailed description of prediction score in the method section at **lines 397-407.**

10. Lines 77-79: How are the numbers of S genes and R genes determined? Please provide clarification on this.

In the revised manuscript, we have included the procedure of determining R and S genes in the method section at **lines 403-407**.

11. In Fig. 1a, d, SFig. 1e, g, and relevant main text, it's important to specify the sample size (and may also the p values and 95% CIs) for each AUC calculation.

We have added the sample sizes in the corresponding figure legends.

12. In SFig. 1b and c, the color code appears suboptimal as it can be challenging to distinguish SD from PR/CR. Consider improving the color differentiation.

In the revised figures, we have changed the color codes for a clearer contrast.

13. In SFig. 1e, could the authors also compare AUCs between pre- and post-treatment in the original set for additional context?

In the revised figure S1d, we have included the AUCs between pre- and post-treatment in the original set.

14. In SFig. 2a, please display the correlation coefficient and p-value on the figure.

In the revised figures, we have added the correlation coefficients and p-values on the panels.

15. It would be helpful to add the numbers at risk at the bottom of the Kaplan-Meier curves.

In the revised manuscript, we have removed the survival analysis to keep the content focused.

16. Check the consistency between the text (lines 108-111) and the figure panels.

In the revised manuscript, we have corrected the inconsistency issues between the main text and figure panels.

17. In Fig. 2g and h, ensure that the legend and figure are consistent.

In the revised manuscript, we have corrected the inconsistency issue between figure legends and panels.

18. It would be beneficial to include 95% confidence intervals for the hazard ratios (HRs).

In the revised manuscript, we have removed the survival analysis to keep the content focused.

19. At line 80, please ensure that the citation for the figure is accurate.

In the revised manuscript, we have corrected the citation issues.

Reviewer #2 (Remarks to the Author): with expertise in bioinformatics and systems genomics

In this study, Wang and colleagues studied the gene expression difference between immunotherapy responders and non-responders of melanoma patients and identified signatures expressed higher in responders (S signature) and reversely higher in the latter (R signature), and then crossed the signature genes with cancer cell line transcriptomics data treated with chemical compounds or shRNAs. The results validated the immune response relevance and clinical outcome relevance of the signature genes and identified compounds that may enhance anti-PD-1 immune therapy. Very importantly, the authors further performed cell line and mouse study to validate the strong candidates and to elucidate the potential mechanism. The study approaches and results are both very interesting and comprehensive, with strong bioinformatics analysis and experimental validation. The manuscript is overall very well written and presented clearly, except a few places. The work could be significant to the field for developing combined treatment. I don't have much more to contribute to improve the manuscript significantly, because it is already

excellent. Here are a few comments that the authors may evaluate if they are helpful.

We appreciate the reviewer's positive evaluations and very helpful comments. In the revised manuscript, we have followed the reviewer suggestions to improve the study.

1. It would be interesting to know how many of the R/S signature genes are expressed in different TME cell types highly or specifically, particularly immune cells, like macrophages and NK cells. This is relevant, because the observed gene expression difference between patients may reflect the cell composition change in their TME.

We agree with the reviewer that this is an important question to address in most of the bulk RNA-seq based immunotherapy outcome predictions. We have analyzed single cell RNA-seq data (GSE115978) from melanoma patients treated with anti-PD-1 therapy. For genes in R signature, we observed that more than 74% of them are expressed in tumor cells, and approximately 50% of them also show expression in cancer-associated fibroblast, endothelial cells, as well as macrophages. For genes in S signature, we found quite some of them showed expression in macrophages, followed by T cells NK cells and tumor cells. Given the limited availability of single cell RNA-seq data from paired pre- and post-treatment samples, at current stage, it is hard to measure the exact treatment-induced expression changes in each cell type. However, since the bulk tumor RNA-seq usually requires a high tumor purity during sample preparation^{2, 3}, we believed the majority of treatment-induced expression changes observed in bulk tumor samples comes from the tumor cells. In the future, when paired pre- and post-treatment samples for single cell RNA-seq become available, we will be able to expand our analysis to genes that reflect treatment-induced cell composition changes in tumor microenvironment.

2. In related to 1, as transcriptomics from different platforms were used in the studied, some including all human genes other not. Can the 1190 R and 130 S genes be further reduced to smaller core sets that are consistently present across data and remain predictive?

In the revised manuscript, we have updated our procedure with a more robust and unbiased implementation to identify the response signature. The new signatures have a more balanced gene numbers between R (419 genes) and S signatures (366 genes), respectively. The new signatures have a classification performance of 0.94 in cross-validation and 0.80 in leave-out validation (AUC). Detailed results can be found at **lines 81-94** and **Fig. 1c, d and e**. We have also redone all the downstream analysis in terms of the drug prediction and mechanism analysis based on new signatures. We thank the reviewer's suggestion that helps us find a more balanced set of core signatures, which are consistently present across data and remain predictive.

3. The expression for most genes in the CMAP was derived from imputation, inferred from their expression correlation with the landmark genes that were experimentally measured. Is this a concern in the usage of the data for computing scores for R/S genes sets, as genes in the same set are likely correlated?

It appears to us the original signature, especially the R signature (with 1190 R genes), indeed has some redundancy with correlated genes included in the same set. We believe the new core signature has alleviated this concern. We also agree with the reviewer that the imputed gene expression from L1000 platform is the trade-off that many users of CMAP will need to take in exchange for the capacity to in silico screening large number of drugs. Based on the correspondence with the L1000 team and our own experience, we think the quality control of CMAP data are excellent. Moreover, we have performed *in vitro* and *in vivo* experimental validation for the prediction and underlying mechanism. We hope these efforts would help address the reviewer's concern.

4. In a few places, the text may be expanded a little to include more details. This can be achieved and will be helpful to readers. The current journal doesn't have word limitation. For examples, the first line in Result, "We obtained the paired transcriptomic data ...". The authors can simply clarify if the data is from blood or tumors, even though the information is in M&M. Following this, "PCA revealed that the anti-PD-1 induced expression achieved a better classification performance (AUC=0.77)..." This statement seems to condense multiple analyses into one; PCA analysis of what? How does PCA connect to AUC? Your readers are not necessary all computation researchers.

We thank the reviewer for pointing this out. We have added brief description in the revised manuscript (**lines 74-75**) for the tumor sample information. For the PCA analysis, we have also included a brief description in the main text and figure legend. Specifically, we used the first principal component of pre- or post- treatment gene expression to classify the patient response to anti-PD-1 therapy. The AUC is obtained by defining responders and non-responders based on different cutoffs of the first principal component. By comparing prediction with the clinically defined responders and non-responders, we will be able to calculate the false positive rate and true positive rate, by which we can calculate the AUC.

5. The authors found that post-treatment transcriptomics data are better for prediction than pre-treatment ones. So, how many of the R/S genes showed differential expression before and after anti-PD-1 treatment? As the goal for clinical usage, we probably want to consider pre-treatment data. Most of the other gene expression data analyzed in the manuscript are not from post anti-PD-1 treatment either. Would excluding the differentially expressed genes before and after treatment from the R/S signature genes have any impacts on the results and key conclusions? In related to this point and #3 above, please clearly indicate if data is from patients before or after treatment in the text when patient data are described in each analysis.

We agree with the reviewer that our study is not intended to identify a predictive signature that can be used in clinical situations to predict individual patient response. Instead, the signature identified in our study served as a tool to find promising drugs that are synergistic with anti-PD-1 therapy. The treatment-induced signature can be relatively easier applied to the L1000 data and identify which chemotherapy compounds induce similar signature gene expression to shift the anti-PD-1 response. We have included this discussion in the revised manuscript (**lines 279-290**).

To further address the reviewer's comments, we have included two more datasets that have paired samples before and after anti-PD-1 therapy during the validation procedure. We validated our signature in these paired datasets, and we found our signature showed generally better performance in paired or post-treatment samples compared to pre-treatment ones. We have added the treatment information for every dataset we used in the study. Related results can be found in **Fig. 1** and **Extended Data Fig. 1**. We have also analyzed the differential expression of R and S genes before and after treatment. We found most of the R genes are expressed higher in before-treatment samples, whereas S genes express higher in after-treatment samples. Related results can be found in **Extended Data Fig. 1e**.

6. In evaluating the prediction performance, do some patient (or cell line) samples contain replicates? How much variation exists for the results between replicate samples, e.g., R-to-S shift?

Regarding the cell line samples, the shRNA and compound data we have used are CMAP level 5 signatures, which have already been normalized based on technical replicates. In addition, we have done a pre-filtering on the shRNA and compound signatures before calculating the R-to-S shifting score. We only chose the shRNA and compound experiments that show a relatively high transcriptional activity to make sure the chosen experiments have consistent gene expression profiles among replicates. We have evaluated the variance among shRNAs targeting the same

genes, and we found 77% of them showed a low standard deviation. For the compound experiments, most of the compounds have been tested at different dosages and different time points in the same cell lines. However, shift ability can vary across treatment time and dosage. For patient samples, multiple biopsies from the same samples are grouped into one sample by taking the average during the analysis.

Reviewer #3 (Remarks to the Author): with expertise in cancer immunotherapy

In this study, Dr. Yang and colleagues analyzed a number of public datasets to identify the gene expression signature that may predict tumor sensitivity to anti-PD1 therapy, and, further, they showed that perturbation of resistant genes by chemotherapy can cause a shift to the sensitive gene signature. Mitophagy pathway is proposed as a potential target for the combination of chemotherapy and immunotherapy. However, there is a lack of sufficient experimental evidence to support the strong claim of chemo-immunotherapy synergism.

1. In Fig. 1, the author begun with a computational analysis of paired transcriptomic data from melanoma specimens, but then jumped to the analysis of multiple types of cancers, in which the transcriptomic data are probably not paired. The connection is not quite clear to me. Why in Fig. 1a PR and CR are combined for the analysis, while they are separated in Fig. 1c? Average gene expression doesn't show a pervasive difference between PR and PD, in terms of either R signature or S signature.

In the revised manuscript, we have incorporated a bootstrapping resampling and cross-validation procedure (**lines 81-84, Fig. 1, Extended Data Fig. 1, and method lines 372-412**) to identify more compact anti-PD-1 response signatures. With the new signatures, we have validated their performance in two additional melanoma patient cohorts with paired samples (**line 81-94 and Fig. 1c-e**). We have included an additional melanoma mouse model (B16) and cell lines (i.e., MEL888 and MEL526) to validate our synergism prediction and mechanism (**Fig. 3, 5, and 6**). We hope the reviewer would agree with us that our computational framework is working if provided with good quality data in the same cancer type (e.g., melanoma).

We indeed have some additional discoveries that the R/S signatures are also predictive to other cancer types in both after-treatment patients and TCGA patient cohorts. We speculated that this is because the signatures capture some shared mechanism (e.g., mitochondria damage induced type I interferon activation) in multiple cancer types. We have added a discussion and urged for careful interpretation of our computational prediction (**line 319-326**).

To keep consistency throughout our analysis, in the revised manuscript, we have combined patients with PR and CR for all the datasets used in the study. We would like to thank the review's comments, which have significantly improved the rigorousness of our study.

2. The inverse correlation between T cell infiltration and R signature is not strong in cancer types other than SKCM in Fig. 2d. It raises a question whether the finding should be only applied to certain cancer types, such as melanoma, rather any cancer types.

We agreed that at the current stage, our signature may be most predictive to melanoma. In the future, when more paired data (pre- and post-treatment) are available for other cancer types, we will be able to construct cancer-specific signatures and design cancer-specific synergy screening procedure. We have included a discussion on this limitation in the discussion section (**lines 319-326**).

3. Different types of cancers have diverse transcriptomics and distinct tumor microenvironment, so it is surprising to me that perturbation of a single given R gene or S gene can cause a strong signature shift in many cancer types.

Based our pathway analysis on R and S signatures, it appears to us that these specific set of genes may recapitulate some shared mechanism of immunotherapy response such as type I interferon response of the cancer cells triggered by treatment. To further address the reviewer's concern about the multiple cancer type analysis, we have included three animal models (i.e., B16, CT26, and MyC-CaP) and more cancer cell lines (i.e., A549, PC3, and HT-29) to validate our discoveries. These results can be found in **Fig. 3, 5, 6** and **Extended Data Fig. 3, 5, 6**.

4. When intersecting the knockdown analysis with drug treatment analysis, with regard to shift ability, how is the correlation or reproducibility between these two kinds of treatments?

In the revised manuscript, we have now included the intersected results in shRNA knockdown and drug treatment analysis (**Fig. 4** and **Line 184-196**). The PAKi drug that is picked for further experimental validation is the one reproducibly identified in both genetic and pharmacological inhibition.

5. The data from CT26 model in this study does not support a strong claim of synergism, because either anti-PD1 or dox alone already inhibits tumor growth a lot. The TIL analysis should be performed with four groups of treatments as in Fig. 5d to show the immunological basis of synergism.

Thanks for the reviewer's suggestion. In our original submission, the dosage of doxorubicin we gave to the CT26 mouse was too high that doxorubicin alone has a strong antitumor effect. In the revised manuscript, we have redesigned and redone the experiments in CT26 (colon carcinoma) and B16 (melanoma) syngeneic mouse tumor models. The TIL analysis is now performed with four groups of treatments to show the immunological basis of synergism. The results indicated that doxorubicin + anti-PD-1 treatment induced synergistic tumor suppressive effect in tumor growth (**Fig. 3e, f**). Additional flow cytometry analysis in all four different groups revealed the synergistic alterations in tumor-infiltrating lymphocytes (**Fig. 3g, h** and **Extended Data Fig. 3d-h**).

6. In Fig. 6, two new cell lines were introduced for investigation of mitophagy. Why the cell lines in the previous figures were not used here to make data consistent and comparable?

In the revised manuscript, we have now included data from A549, PC3, HT-29 cell lines to validate our discoveries. All these cell lines were included in the previous computational analysis figures. The result can be found in **Fig. 5j**, **Extended Data Fig. 5 c, e-g, 6f, g** and **Table S8, S10**.

7. There seems to be a technical issue with the immunoblots of phospho-STING in Fig. S6f and g.

All Phospho-STING immunoblots were repeated with a new phospho-STING antibody (CST, #50907). The result can be found in **Extended Data Fig. 6a, b**. The original uncropped immunoblots of phospho-STING and STING are given below.

8. Mitophagy removes the damaged or dysfunctional mitochondria, which is a key source of intracellular dsRNAs. How could the enhanced mitophagy by PAK inhibition generate more dsRNAs? Are other pathways also involved when targeting PAK? Does PAK inhibitors sensitize refractory tumors to anti-PD1 in syngeneic tumor models? *In vivo* immunological analysis should also be performed.

In the revised manuscript, we performed mitophagy detection assay to demonstrate that doxorubicin and PAKi (PF-03758309) can significantly increase mitophagy (**Fig. 5d, c** and **Extended Data Fig. 5c-f**). We have also specifically detected the mitochondrial RNA (mtRNA) in cytoplasm and demonstrated that PAKi can significantly increase cytosolic mtRNA (**Fig. 6b, c** and **Extended Data Fig. 6e**).

We tried to use two different autophagy inhibitors (DC-LCin-D5: an LC3B inhibitor and Chloroquine: an inhibitor of lysosome-autophagosome fusion) to inhibit mitophagy. The result is not conclusive whether blocking autophagy/mitophagy can rescue PAKi-mediated type I interferon gene expression.

With these observations, we have now revised our conclusion to drug-induced mitochondria damage releases mtRNA and activates type I interferon signaling. We sincerely thank the reviewer's insightful comments. In the future work, we will continue to study how PAK inhibition led to increased mtRNA cytoplasmic release and if there are other pathways also involved when targeting PAK.

We have also tried to study the synergistic effect of PAKi (i.e., PF-03758309) with anti-PD-1 using *in vivo* syngeneic TNBC tumor model (4T1.2). As shown in two batches of animal data below, we found that PF-03758309 showed very high toxicity in mice. In mouse models treated with different dosages, PF-03758309 treatment leads to dramatic weight loss and poor body composition. In the future, we will work with medicinal chemists to optimize its toxicity and PK for *in vivo* study.

Reviewer #4 (Remarks to the Author): with expertise in computational biology, oncology

The authors compare gene expression of paired pre- and on-treatment patient transcriptomics data to identify gene signatures for sensitivity and resistance to anti PD-1 therapy. Using these signatures, they introduce a shift ability score to assess immune response of chemo-/targeted therapy and link it to the potential synergistic effect of anti PD-1 therapy. They perform multiple *in silico* and *in vivo* experiments to show the validity of their approach. They characterized biological mechanisms that potentially can increase the efficacy of PD-1 immunotherapy treatment when used in combination with chemo-/targeted therapy. Identification and experimental validation of mitophagy as immunotherapy and chemo-/targeted therapy synergism mechanism is an important finding that further investigated and translated to clinical practice can benefit patients. However, the manuscript has several shortcomings, please see below.

Comments:

1. Authors write about chemo-immunotherapy combinations and synergies, however the term “immunotherapy” in the manuscript is limited to the single PD-1 inhibitor (nivolumab) and the term “chemotherapy” seems to include kinase inhibitors (e.g. MEK, RAF inhibitors). I suggest treating both terms more precisely.

We thank the reviewer for this comment, which helped us to improve the rigorousness of our study. In the revised manuscript, we have changed the “immunotherapy” to “immune checkpoint blockade”, and we have specified in the text that the “chemotherapy” in our study included both conventional cytotoxic chemotherapy agents and small molecular inhibitors for targeted therapy.

2. The Introduction section gives a very high-level overview of the combinatorial cancer treatment and interaction between immunotherapy and chemo-/targeted therapy accessible to the general audience, whilst graces literature related to the main section. For example, no mention of the PD-1 function and inhibitor (a single immunotherapy considered in analysis) is available. Desirable would be a structured overview of related work in terms of application and methodological domain.

In the revised introduction paragraph, we have included an overview of anti-PD-1 centric studies. We thank the reviewer for helping us improve the readability of the manuscript.

3. The R and S signatures were determined based on the 43 patients only. Do the authors consider their approach to be statistically robust? For example, bootstrapping or cross-validation may assess this. Including 1,329 genes in R and S signatures may also include many false positives and noise. Also, a power analysis may be beneficial.

We thank the reviewer for this insightful suggestion. In our revised manuscript, we have incorporated the bootstrapping resampling and cross-validation during the signature construction procedure. The obtained signatures achieved an average AUC of 0.98 in cross-validation and 0.80 in leave-out validation. The above approach has also refined our new signatures into more balanced gene numbers, with 419 R genes and 366 S genes, respectively. Details of the related results can be found at **lines 81-94, Fig. 1, Extended Data Fig. 1, and method lines 372-412.**

We have also identified two independent patient cohorts and tried to merge the three datasets to increase the sample size and statistical power. However, the batched effect in the different pre- and post- treatment datasets appears to be very difficult to deal with (Please also see our response in comment #6). In this regard, we decided to use the original dataset as training set and two additional patient cohorts as validation datasets.

4. What is the main reason for using TCGA data for validation? It does not assist in anti PD-1 therapy and synergistic effect evaluation. Also, the authors write: “Although TCGA patients did not receive immunotherapy, they have undergone intrinsic immune response processes which lead to the infiltration of immune cells into the tumor microenvironments.” It is not clear why they set it similar to the anti PD-1 therapy. I assume that also in immunotherapy, intrinsic immune responses take place. To reduce the results overload excluding analysis on the TCGA data or replacing it using data from Gene Expression Omnibus (GEO) can be considered.

In the revised manuscript, we have now included two more independent cohorts (MGH cohort and PRJEB23709) in the training and validation procedure. We have also moved the majority of TCGA results to the supplementary.

5. The authors write that “combining 181 the compound and shRNA screening result revealed 49 drug targets who’s genetic and 182 pharmacological inhibitions can induce R-to-S shifting in the same cell lines.” They provide some examples of those targets, e.g. BRAF 28, RRM1 10,11, CDK129, CDK430, HDAC123, TOP1 as well as PAK4 (described as novel target). They further experimentally validated in vivo the synergistic effect of anti PD-1 therapy and doxorubicin.

However, this effect has been already described in the literature. What is the rationale behind the choice of target to validate experimentally? And is there any novel, previously undescribed drug targets discovered?

We chose these two compounds to validate because of their translational potential. Doxorubicin is an FDA approved drug that can have immediate clinical impact for the patient. PAKi is a promising compound that serves as a good lead compound for new drug development in the long term.

In addition to performing the experiments that validate the doxorubicin and PAKi's synergism with anti-PD-1, our computational analysis also revealed that the underlying mechanism of this synergism is the mtRNA-induced type I interferon activation due to mitochondria damage. In the revised manuscript, we have experimentally validated this mechanism for both doxorubicin and PAKi in multiple cancer models. As shown in **Fig. 4a**, there are quite some novel, previously undescribed drug targets discovered for future validation. The completed list of identified drug targets are available in **Table S6**.

6. The authors use three different datasets for their analysis. Why did they decide not to harmonize the transcriptomic data coming from different sources (e.g. batch correction, tissue correction). However, if they did, it would be great to provide the details on the used approach.

We thank the reviewer for this insightful suggestion. We have collected another two independent studies which have paired pre- and post-anti-PD-1-treatment samples (the MGH cohort and PRJEB23709). We noticed that different studies, although all of them are anti-PD-1 treated cohorts, could have undergo a totally different sample collection, sequencing, and response evaluation protocols, which will lead to significant batch effects. We have tried to remove the batch effects among studies, but we found this will also lead to a significant loss of real signals and produce many false positive results.

On the other hand, since studies with paired samples are still very limited (in most of the cases, only pre-treatment samples are available), it is hard to find other studies serving as independent validation cohorts. With these considerations, we decided to use the aforementioned datasets as validation cohort, instead of harmonizing them into one merged training set.

Related results can be found at **lines 81-94, fig. 1, Extended Data Fig. 1, and method lines 338-418**.

7. The interpretation of the introduced shift ability score (line 421) is not completely clear since the enrichment score (for both ES_R and ES_s) covers genes that are over-represented at either the top or bottom of the ranked list. Could you please elaborate more on the interpretation of shift ability score (lines 422-427)?

In the revised manuscript, we have included a detailed description of the enrichment method used in our study in the method section (**lines 430-443**). Briefly, we implemented the GSEA pre-rank module, which evaluates whether genes from a gene set are either at the top or at the bottom of the ranked list.

In the revised manuscript, we further improved our calculation of shift ability score by using the normalized enrichment score instead of the raw enrichment score. The normalized score is calculated based on the permutation of the ranked list, providing a more unbiased score compared to the raw one. The normalized enrichment score is directional (which is the same as the raw enrichment score): a positive value usually means genes from the given gene set are more enriched at the top, whereas a negative value usually means genes are more enriched at the bottom.

8. The code repository is very unstructured. It further does not contain any instructions on how to set up a working environment and run the code. With the initial code state, the analysis results can be barely reproduced.

The performed analysis workflow is separated into different subsections (e.g. Predicting anti-PD-1 response using patient gene expression profiles or Construction of anti-PD-1 response signature based on treatment-induced expression, etc.). However, it still remains very unstructured, the steps when and after the shift scores are calculated are not easy to understand. I suggest introducing more structure into the methods (and consequently results section) including clearly indicating the main goal of this analysis parts.

We thank the reviewer for this insightful suggestion. We have re-organized the code and annotated each critical step, e.g., the training and construction of R and S signature, the calculation of shift ability score. The updated code repository can be accessed at: <https://github.com/DaYangLab2015/ChemolmmunoSyng>.

9. Also, the current manuscript contains a lot of results which are not very critical for the main contribution in identifying potential synergistic targets. For instance,

Classification in Treatment-induced expression change profiling can predict anti-PD-1 response in patients subsection (lines 71-76)

All validation experiments on GSE93157, GSE168204 and some experiments on the TCGA dataset (lines 80-91, and more)

Survival analysis (lines 112-118) Etc.

Certainly, the experiments are of high importance for justification and validation of every analysis step, they impede the understanding of the main manuscript message. The authors can consider moving the external validation analysis and corresponding figures in the supplementary materials. This would improve the readability and understanding of the manuscript significantly.

We agree with the reviewer that the signature is not for individual patient response prediction, but rather identification of the potential synergistic targets. We have moved the majority of the TCGA results to the supplementary data. We have also reorganized the paper and emphasized our focus on identifying synergistic drug combinations instead of predicting anti-PD-1 responses in patients.

10. The approach to calculate R and S signatures and use it for novel target discovery can be potentially interesting for other applications. What do authors think about it? Or is there anything that limits its applicability only to PD-1 combination therapy? What could be the follow-up or future work? Please, address it in the discussion section.

We thank the reviewer for this insightful suggestion. We have included a paragraph to address these questions in the discussion section (**lines 299-316**).

Minor comments:

12. In the introduction section, no references are provided for multiple statements making the writing sound as an opinion. Below are only examples, I recommend the authors to revise their citations:

-) “Even for the patients who initially respond to the therapy, the later developed immunotherapy resistance remains to be challenging” (lines 37-38)

-) “However, the design of the combination regimens so far is largely relied on clinical experiences, it is very challenging to characterize new chemo-immunotherapy synergisms. The emerging large-scale pharmacological transcriptomic datasets that profile the expression changes after

drug/immunotherapy treatment provide us deeper and novel insights on how treatment changes biological processes in the tumor.” (lines 52-55)

-)“Of note, by 2023, FDA have approved several chemo-immunotherapy regimens in diffuse large B-cell lymphoma (Polatuzumab + bendamustine/rituximab), triple-negative breast cancer (Atezolizumab/Pembrolizumab + taxanes), gastric cancer and esophageal adenocarcinoma (Nivolumab + FU-/platinum).” (lines 44-48)

In the revised manuscript, we have added the citations for the above statements.

13. Why first infer genes & increase dimension to reduce it in the analysis later again? (Lines 33 -339) How do the authors define best inferred genes? How is the inference quality assessed? We are sorry for the confusion. The post-shRNA/compound-treatment transcriptome data used in this study is obtained from Connectivity Map (CMAP) database⁴. We did not infer the genes by ourselves, instead, the CMAP database (which is based on L1000 platform) has their own protocol on defining the landmark genes and best inferred genes. Briefly, the L1000 platform only detects the gene expression of 978 landmark genes. These landmark genes will be further used as model input for the expression estimation of other genes. According to the CMAP publication, “...9,196 of the 11,350 inferred genes (81%) correlated with p-value less than or equal to 0.05. This set of 9,196 inferred genes, plus the 978 landmarks, are referred to as the Best Inferred Genes (BING)”. Details can be found in the attached reference⁴.

14. The authors use different numbers of melanoma patients in the GSE91061 dataset used to determine S- and R-signatures. Please consider updating the inconsistent numbers:

-) 42 patients used for the analysis (lines 343-347)

-) 68 patients in the results section (line 71)

-) 41 patients in Figure S1a

In the revised manuscript, we have corrected the inconsistency number issues.

15. Authors use “gene expression” as synonym to “gene expression changes” in multiple occurrences, when describing the “log₂-transformed fold change between on- treatment and pre-treatment expression”. For example,

-) “ In 9,626 patients across 23 cancer types from TCGA database, we observed a dramatically negative correlation between expression of R genes and S genes...”

-) Specifically, we observed that S signature expression are strongly correlated with immune-hot phenotypes...” (line 96)

-) “Increased expression of R signature is highly correlated with decreased infiltration of CD8+ T cells...” (line 98)

-) “Y-axis represents the Wilcoxon rank-sum test of gene expression on treatment-induced level between anti-PD-1 sensitive patients and resistant patients.” (lines 745 - 747, Figure 1b)

In the revised manuscript, we have corrected the inconsistency terminology issues.

16. It is not clear what PHS001919 (line 82) refers to. It is not described in the materials and methods section.

In the revised manuscript, we have added the detailed description of data we used in the method section.

17. Some terms are not defined before the first usage. For example, the authors do not explicitly explain how they map PD, ST, PRCR response scores (lines 342-354) to responders and non-responders (368-369, results section and figures)

In the revised manuscript, we have included the definition of responders and non-responders in the **Supplementary Table 1** and **method** section.

18. In the result section, the authors may have incorrect references to the subfigures of Figure 1:

-) “Fig. 1a, b...” line 75 should be Fig. 1a

-) “Fig. 1c” line 80 should be Fig. 1b

In the revised manuscript, we have corrected the figure citation issues.

19. Figure. It is not clear what “in different timepoints” in the sentence “ROC curve shows the classification performance in different time points.” refers to. (lines 772-773, Figure S1e)

In the revised manuscript, we have replaced the “different timepoints” with “pre-” and “post-” treatment.

20. There might be a contradiction in the last paragraph of the discussion section. What is their conclusion?

-) “The mitophagy-mediated release of mtDNA and mtRNAs activates the anti-viral signaling, which will initiate the innate immune response and induce the CXCL10 secretion. Both innate immune response and CXCL10 secretion have been demonstrated to increase the efficacy of immune checkpoint blockades”

-) “Future clinical studies are warranted to determine whether drug-induced mitophagy should be minimized to control chemo-resistance or be exploited to synergize immunotherapy.”

We are sorry for the confusion. We have now slightly changed our conclusion to drug-induced mitochondria damage releases mtRNA and activates type I interferon signaling. The mitophagy we observed could be an indication of mitochondrial damage. We have revised the discussion accordingly.

21. The authors denote classification models with AUC between 0.66 and 0.75 as “robust”. (Lines 87-88). In my opinion, robust models are defined differently and are comprehensively tested.

In the revised manuscript, we have modified our description on the signature performance.

22. The paper concludes that “Future clinical studies are warranted to determine whether drug-induced mitophagy should be minimized to control chemo-resistance or be exploited to synergize immunotherapy.” It is not clear why authors refer directly to clinical trials and skip in vitro and in vivo validation of the drug-induced mitophagy for immunotherapy and chemo-/targeted therapy synergism.

In the revised manuscript, we have modified the related statements in discussion.

Reference

1. Seymour L, *et al.* iRECIST: guidelines for response criteria for use in trials testing immunotherapeutics. *The Lancet Oncology* **18**, e143-e152 (2017).
2. Bell D, *et al.* Integrated genomic analyses of ovarian carcinoma. *Nature* **474**, 609-615 (2011).
3. Program TCGA. <https://www.cancer.gov/ccg/research/genome-sequencing/tcga>. (ed[^](eds)).
4. Subramanian A, *et al.* A Next Generation Connectivity Map: L1000 Platform and the First 1,000,000 Profiles. *Cell* **171**, 1437-1452 e1417 (2017).

REVIEWERS' COMMENTS

Reviewer #1 (Remarks to the Author):

The authors have made effort to address most of my concerns. However, I still do not quite understand why "Interestingly, we identified that most of the immune response happened in SD patients instead of PR/CR patients". To my understanding, no matter SD is classified into responders or non-responders, PR/CR should always have stronger immune response signal than SD. I do not have any further comments.

Reviewer #2 (Remarks to the Author):

The authors have significantly improved their manuscript in this revision and provided reasonable responses to my comments. It's not essential, but it'd be good to mention what %s of the R and S signature genes showed high expression in stromal and immune cells in the tumors. Authors described this in their response letter. This information may be included in M&M.

Reviewer #2 (Remarks on code availability):

The codes are there. I reviewed the codes but didn't download and test them. The authors provided the main codes for the model building and screening but not the codes for individual figures, as described in their Code Availability.

Reviewer #3 (Remarks to the Author):

The revised manuscript is significantly improved. The authors have provided new evidence to support their conclusions. A few questions remain below.

Only a limited number of overlapped hits between genetic knockdown and chemical inhibition were found. Should one expect a considerable portion of common hits?

Low dose doxorubicin and anti-PD-1 demonstrated synergism in suppressing tumor growth and elevating CD4+IFN γ + T cells. Did authors look at the abundance and phenotype of CD8+

T cells, which are considered as the prominent responders to anti-PD-1 treatment? Which PD-1 antibodies did the authors use for blockade and for flow cytometry analysis? Did the PD-1 blocking antibodies interfere with the flow cytometry detection of PD1 expression?

For the phospho-STING detection in the original uncropped images, why did the cell lines with high STING expression give low phospho-STING signals, while those with very low STING expression show relatively high phospho-STING signals?

Reviewer #3 (Remarks on code availability):

I am not a bioinformatic expert.

Reviewer #4 (Remarks to the Author):

Thanks a lot for the edits, all comments have been comprehensively addressed.

Detailed Point-by-point response to REVIEWER COMMENTS

Reviewer's Comments:

Reviewer #1 (Remarks to the Author)

The authors have made effort to address most of my concerns. However, I still do not quite understand why "Interestingly, we identified that most of the immune response happened in SD patients instead of PR/CR patients". To my understanding, no matter SD is classified into responders or non-responders, PR/CR should always have stronger immune response signal than SD. I do not have any further comments.

We apologize for the confusion in language. We agree with the reviewer that the PR/CR should always show the highest immune response signal, and the SD is an intermediate stage between PD and PR/CR. In the response letter we meant to say "most of the immune response started to happen in SD tumors, even though the tumor sizes haven't shown big changes. This is indicated by the enrichment of immune activity we observed in SD tumors". We thank the reviewer for pointing this out.

Reviewer #2 (Remarks to the Author)

The authors have significantly improved their manuscript in this revision and provided reasonable responses to my comments. It's not essential, but it'd be good to mention what %s of the R and S signature genes showed high expression in stromal and immune cells in the tumors. Authors described this in their response letter. This information may be included in M&M.

We thank the reviewer for the suggestion. We will include this information in the M&M.

Reviewer #3 (Remarks to the Author)

The revised manuscript is significantly improved. The authors have provided new evidence to support their conclusions. A few questions remain below.

Only a limited number of overlapped hits between genetic knockdown and chemical inhibition were found. Should one expect a considerable portion of common hits?

We think the limited number of overlapped hits have three reasons: (1) the number of overlaps between the shRNA and chemical inhibitor targets are not high in original dataset; (2) both shRNA and small molecular inhibitors are known to have off-target effects; and (3) the efficacy of chemical inhibition can vary largely depends on different dosages and treatment durations. In fact, these differences between shRNA and drug treatment are the major rationales for us to overlap the results from genetic and pharmacological screening in our study. We thank the reviewer for this insightful comment.

Low dose doxorubicin and anti-PD-1 demonstrated synergism in suppressing tumor growth and elevating CD4+IFN γ + T cells. Did authors look at the abundance and phenotype of CD8+ T cells, which are considered as the prominent responders to anti-PD-1 treatment? Which PD-1 antibodies did the authors use for blockade and for flow cytometry analysis? Did the PD-1 blocking antibodies interfere with the flow cytometry detection of PD1 expression?

We did check the abundance of CD8+ T cells, but the numbers of CD8+ T were very low in B16 model when we took the tumor for immune profiling. This observation is consistent with published results (*Bioelectrochemistry*, 2021, 140, 107831). We believe the strong synergism can be indicated by the percentages of CD4+IFN γ + and CD4+PD-1+ (Figure 3g). As for the results in the MyC-CaP model, we can observe significant induced infiltration of CD8+ T cells in both anti-PD1 and combination treatment. Comparing with the Dox treatment alone, the combination treatment significantly decreases the CD8+

PD1⁺ cells ($P < 0.0001$, **Figure 1**). The combination treatment showed a similar but nonsignificant trend of decreased CD8⁺PD1⁺ T cells compared with anti-PD1 alone.

Figure1. Infiltrated CD8⁺PD-1⁺ T cells in MyC-CaP prostate cancer following different treatments. Antibody for blockade is BioXCell InVivoMAb anti-mouse PD-1 (CD279) Catalog #BE0146. Antibody for flow study is Brilliant Violet 615 anti-mouse CD279 (PD-1) Antibody (BD Biosciences, Cat# 752354). We don't think the PD-1 blocking antibodies would interfere with the flow cytometry detection of PD1 expression. As we prepared the single cell suspension, the PD-1 blocking antibodies should be removed with the supernatant when we processed the cell suspension.

For the phospho-STING detection in the original uncropped images, why did the cell lines with high STING expression give low phospho-STING signals, while those with very low STING expression show relatively high phospho-STING signals?

- Cell lines with high STING expression give low phospho-STING signals: Activation level of phospho-STING could be low in those cells.
- Cell lines with low STING expression show relatively high phospho-STING signals: We would like to clarify that the blot exposure duration is different for Phospho-STING (i.e., ~8 min) and STING (i.e., ~2 min) in the uncropped images. We provided long exposure time images to show reviewers a complete band profile and that PAKi did not activate STING. Overall, different cell lines exhibited different endogenous STING expression levels. Since the STING signaling-based results did not provide major support for the PAKi mechanism, we did not perform in-depth analyzes of variable expression of STING between cell lines. We thank the reviewer for this insightful comment.

Reviewer #4 (Remarks to the Author)

Thanks a lot for the edits, all comments have been comprehensively addressed. We thank the reviewer for helping us improve our manuscript.